# Experimental evidence for circular inference in schizophrenia

Renaud Jardri[1,2], Sandrine Duverne[1], Alexandra S. Litvinova[3] & Sophie Denève[1]

Schizophrenia (SCZ) is a complex mental disorder that may result in some combination of hallucinations, delusions and disorganized thinking. Here SCZ patients and healthy controls (CTLs) report their level of confidence on a forced-choice task that manipulated the strength of sensory evidence and prior information. Neither group's responses can be explained by simple Bayesian inference. Rather, individual responses are best captured by a model with different degrees of circular inference. Circular inference refers to a corruption of sensory data by prior information and vice versa, leading us to 'see what we expect' (through descending loops), to 'expect what we see' (through ascending loops) or both. Ascending loops are stronger for SCZ than CTLs and correlate with the severity of positive symptoms. Descending loops correlate with the severity of negative symptoms. Both loops correlate with disorganized symptoms. The findings suggest that circular inference might mediate the clinical manifestations of SCZ.

[1] École Normale Supérieure, Institut de Sciences Cognitives, LNC (INSERM U960), Paris F-75005, France. [2] Université. de Lille, SCALab (CNRS UMR9193) & CHU de Lille, Hôpital Fontan, Pôle de Psychiatrie (CURE), Lille F-59037, France. [3] Faculty of Biology, Lomonosov Moscow State University, 119991 Moscow, Russia. Correspondence and requests for materials should be addressed to R.J. (email: renaud.jardri@chru-lille.fr).

Schizophrenia (SCZ) is a serious and disruptive mental disorder that has a massive effect on public health[1]. Approximately 51 million people worldwide satisfy the diagnostic criteria for SCZ, but the great heterogeneity in clinical presentation has led some authors to break down SCZ into more limited symptom clusters[2–4]. At the clinical level, psychotic, negative, disorganized and affective dimensions have been considered[5]. Despite intensive investigation, the neural basis of SCZ remains largely unknown. SCZ does not appear to involve focal brain lesions nor is it exclusively related to a particular neurotransmission system. In fact, although five decades of research has provided indirect pharmacological support for dopamine dysfunction in SCZ, virtually all other neuromodulation systems have also been shown to be involved[6,7]. Moreover, SCZ not only affects relatively 'high-level' functions such as social behaviour and speech perception but also 'low-level' mechanisms such as simple perceptual illusions, centre-surround integration and motor adaptation[8,9]. This substantial body of empirical data led to the reconceptualization of SCZ as an equifinal entity[10] (that is, a state of dysfunction that can arise from a variety of aetiological dysfunctions that similarly affect global circuit functions)[11]. Within recently advanced frameworks, the impairment of the regulation of the excitatory/inhibitory (E/I) balance figures prominently[12]. E/I balance is tightly maintained in the mature cortex[13,14], and it clearly plays crucial roles in neural information processing.

On a normative side, the impairments associated with SCZ described above might be understood within the framework of Bayesian inference, predictive coding or both. In this framework, feed-forward and feedback neural processing are interpreted as a propagation of bottom-up sensory evidence and top-down prior knowledge, which are combined using Bayes' theorem. Through this mechanism, the brain is able to form a deep hierarchical representation of the environment, from low-level sensory features to high-level scene interpretations or behavioural choices[15,16]. However, biases in the relative strength of (that is, trust in) sensory evidence versus prior knowledge can lead to the formation of aberrant beliefs (for example, psychotic symptoms)[17–20]. For example, a pathologically high degree of trust in prior beliefs might lead patients to perceive objects that they expect but that are not really there. Similarly, too much trust in sensory evidence can lead to the perception of a non-existent conflict between unreliable sensory information and priors and, eventually, the generation of a false belief system to account for this conflict.

Although these two conceptualizations of SCZ (that is, a 'mechanistic' imbalance in E/I regulation and a 'normative' impairment in Bayesian inference) can account for many phenomena, they have not yet been coherently related. We recently showed how E/I imbalances in hierarchical neural processing can result in a pathological phenomenon called 'circular inference'[21]. Most long-range connections in the human brain are excitatory; the interplay between the feed-forward and feedback connections creates strong recurrent excitatory loops; as a result, the top-down influence of priors on the sensory area can easily be misinterpreted as new sensory evidence and reverberated back up, resulting in 'descending inference loops'. This neural architecture makes us 'see what we expect'. Similarly, sensory evidence might generate high-level representations and be reverberated back down, leading to their misinterpretation as prior knowledge (that is, 'ascending inference loops'). This neural architecture would make us 'expect what we see'. Thankfully, the maintenance of the E/I balance implies that any predictable (redundant) excitatory input is cancelled by inhibition. This predictable input includes reverberated sensory evidence and prior experience. Note that this theory differs from impaired inhibition in predictive coding, in which the top-down messages are prior estimates that inhibit lower levels that transmit predictions errors[22]. In that case, impaired inhibition creates systematic biases in the estimates.

The present study attempts to validate the circular inference hypothesis by quantifying how patients with SCZ and matched healthy controls (CTLs) derive confidence from sensory evidence and prior information as well as how this process is correlated with symptom severity. We used the Fisher task, a variant of the beads task probabilistic paradigm[23,24]. In the beads task, participants must deduce from which of two jars a string of beads has been drawn. Typically, the jars contain beads of two colours in opposite ratios (for example, 8:2 and 2:8). A robust finding in such tasks is that people suffering from psychotic symptoms exhibit a 'jumping-to-conclusions' bias (that is, they make decisions based on less evidence and with increased confidence in their choice)[25–27].

In our variant of the bead task, participants were asked to report the 'chance' (and not an actual decision) that a red fish caught by a fisher came from one of two lakes (left versus right lake)[28]. Each trial started with the presentation of two baskets of different sizes associated with the two lakes. Participants were told to interpret the relative size of the baskets as representing the preference of the fisher for each lake. We interpreted this information as the prior for each lake. The baskets then disappeared and were replaced by two lakes, each containing 50 fishes of two possible colours (red or black), and a fisher between the two lakes holding a red fish (the colour of the caught fish did not change across trials). Each lake contained a different proportion of black and red fishes that varied anti-symmetrically (for example, if the proportion of red fish was 0.1 on one side, then it was 0.9 on the other side), including completely unambiguous trials with proportions of 0:1 and 1:0. This experimental set-up resulted in different quantifiable sensory evidence for the two lakes. The sensory evidence remained available until the response occurred. Importantly, the participants provided a single response regarding their level of confidence about the fish's origins on a scale ranging from totally certain regarding the left lake to totally certain regarding the right lake (rather than signalling a right/left choice; See 'Methods' section and Fig. 1a). Together, these adaptations made our variant of the Fisher task well suited to test the Bayesian inference of patients with SCZ and generate qualitative and quantitative model-based predictions regarding how patients and CTLs integrate prior and sensory evidence to drive probabilistic reasoning. We report converging results indicating that the behaviour of SCZ patients and CTLs could not be explained by simple Bayes but fit well in a parametric circular inference model.

## Results

**Model-free analysis of behavioural performance.** Twenty-five patients with SCZ and 25 matched healthy CTLs performed the task (see 'Methods' section and Table 1 for a full description of the samples). The participants responded by clicking a position on a chance scale that varied from $c = 0$ (absolute certainty for the left lake) to $c = 1$ (absolute certainty for the right lake). The average $c$ was close to 0.5 (equal chance for right and left lake) and did not differ between the groups, excluding a bias for one of the two lakes (Supplementary Fig. 1).

We first used a model-free analysis to confirm that the participants account for the different types of information (that is, fish proportions and basket size) to derive and report a

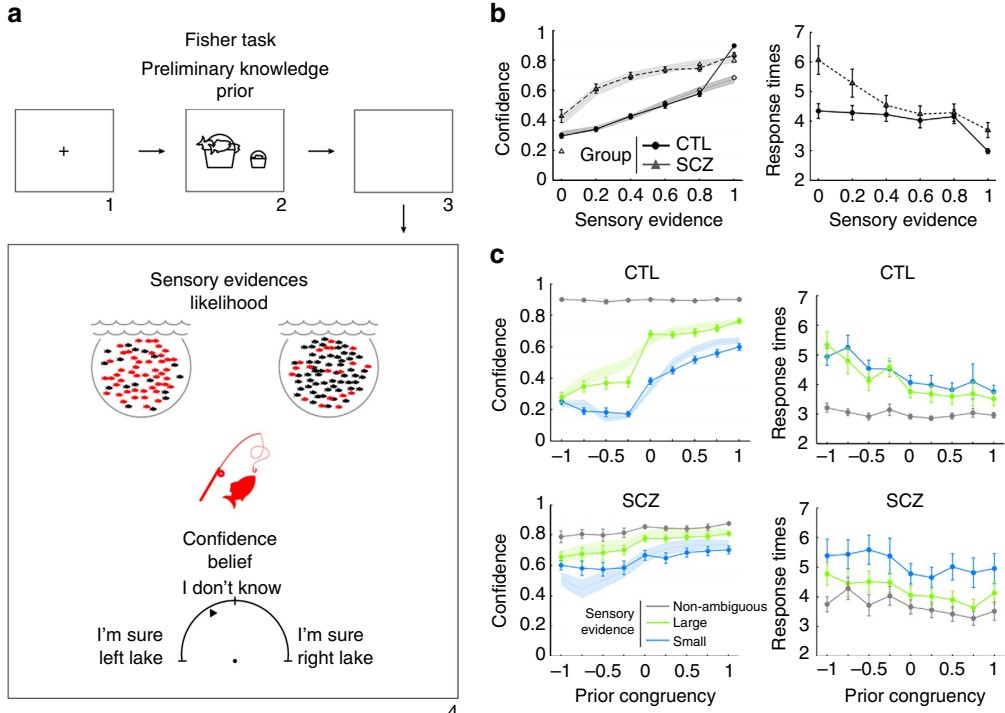

**Figure 1 | The Fisher task: experimental procedure and behavioural performance. (a)** Each trial can be decomposed into four steps (see 'Methods' section). After a fixation cross was presented (1), the prior information was provided (2). The size of the baskets represented the chance that the fisher caught a fish from the left or right lake. This prior was then removed (3), and the likelihood information was provided (4). The likelihood information consisted of the proportions of black and red fishes within both the left and right lakes. The participants reported their confidence that a red fish originated from one of the two lakes using a semi-circular scale. Confidence levels and RTs were recorded. **(b,c)** Mean confidence rates and RTs (in sec) in CTL participants ($n = 25$) and patients with SCZ ($n = 25$). The data are presented as the mean $\pm$ the s.e.m. **(b)** The effects of the sensory evidence on the absolute confidence and RTs in the SCZ and CTL groups. **(c)** The effects of prior congruency on the confidence levels and RTs in the SCZ and CTL groups. Three levels of sensory evidence were plotted: small amount of evidence = likelihood of 0.2 or 0.4; large amount of evidence = likelihood of 0.6 or 0.8; and unambiguous = likelihood of 1. The circular inference model predictions $\pm$ s.d. (estimated with a noise term of s.d. = 0.02 added to the likelihood) are shown as lines with shaded contours (see 'Methods' section and Supplementary Fig. 2 for predictions of the noiseless model for all trials).

graded level of confidence about the origin of the fish. The participant's absolute confidence (defined as $2|c-0.5|$) varied between 0 (completely uncertain; that is, clicking in the middle of the chance scale) and 1 (absolutely certain; that is, clicking at one of the extremes of the chance scale). Sensory evidence was quantified as $2|P_R-0.5|$, where $P_R$ denoted the proportion of red fishes in the right lake. According to the experimental design, the proportion of red fishes in the left lake was always $1-P_R$. Thus, a sensory evidence value of 0 corresponded to a case in which the proportions of the fishes in the two lakes were equal (no sensory evidence), and a sensory evidence value of 1 corresponded a case in which all of the red fishes were found in only one of the lakes (completely unambiguous). Finally, the prior congruency was defined as the basket size for the lake with the largest proportion of red fishes minus the basket size for the opposite lake (set to 0 when the proportions of fishes in both lakes were equal). This difference was scaled to vary from $-1$ (in which the prior contradicted the sensory evidence most strongly) to 1 (in which the prior confirmed the sensory evidence most strongly).

Among the CTLs, the absolute confidence monotonically increased with sensory evidence and increased notably when the sensory evidence was unambiguous (Fig. 1b,c, left panel; main effect of sensory evidence using a general linear mixed model (see 'Methods' section), $F_{1,144} = 14.534$, $P < 0.001$). Moreover, the absolute confidence was substantially lower when the prior was incongruent with sensory evidence (Fig. 1c, left

panel; main effect of prior congruency: $F_{1,328} = 943$, $P < 0.001$), although the effect of prior congruency was strongly modulated by the type of sensory evidence (Fig. 1c, left panel; interaction prior congruency $\times$ sensory evidence: $F_{1,353} = 884$, $P < 0.001$). In trials with unambiguous sensory evidence, the effect of prior congruency was totally abolished ($F < 1$), confirming that the CTLs disregarded prior evidence when the sensory evidence was unambiguous. In trials with ambiguous sensory evidence, the effect of sensory evidence was larger in trials with congruent priors than in trials with incongruent priors (interaction prior congruency $\times$ sensory evidence: $F_{1,312} = 8$, $P = 0.006$). These results confirm that the CTLs accounted for prior evidence when the sensory evidence was ambiguous and revealed that the effect of sensory evidence was steeper when this piece of information was congruent with prior evidence.

In accordance with previous findings[29,30], the patients with SCZ were overall more confident than CTLs (Fig. 1b, left panel; $F_{1,935} = 660$, $P < 0.001$). The conventional indices for jumping to conclusions are provided in Supplementary Fig. 1. Similar to the CTLs, the absolute confidence in patients with SCZ increased with sensory evidence ($F_{1,520} = 319$, $P < 0.001$) and prior congruency ($F_{1,340} = 10$, $P = 0.002$). As Fig. 1 shows, however, these two effects were much smaller in patients with SCZ than CTLs (two-way interactions with group: Fs > 151, ps < 0.001). The significant group $\times$ sensory evidence $\times$ prior congruency three-way interaction ($F_{1,797} = 100$, $P < 0.001$) captured the differential effects of prior congruency between

**Table 1 | Characteristics of the recruited samples.**

|  | SCZ mean ( ± s.d.) | CTL mean ( ± s.d.) | P value |
|---|---|---|---|
| *Demographics* |  |  |  |
| Sample size (*n*) | 25 | 25 | NA |
| Age (y.o.) | 33.2 ( ± 11.3) | 31.2 ( ± 7.5) | 0.3924 |
| Gender (m/f) | 15/10 | 12/13 | 0.3946 |
| Years of education (from first year of primary school) | 11.4 ( ± 0.9) | 11.9 ( ± 1) | 0.1567 |
| *Neuropsychological evaluation* |  |  |  |
| Spatial attention (Bells cancellation task) | 0.5 ( ± 1.5) | 0.6 ( ± 1.6) | 0.9879 |
| Cognitive inhibition (Stroop interference) | 1.5 ( ± 0.8) | 1.4 ( ± 0.5) | 0.1591 |
| Working memory (backward digit-span) | 4.5 ( ± 1.3) | 6.4 ( ± 2.1) | 0.0005* |
| Non clinical beliefs (PDI-21 score) | 10.3 ( ± 6.3) | 2.5 ( ± 2.3) | 2.6e-6* |
| *Illness course, psychopathological evaluation and medication status* |  |  |  |
| Illness duration (years) | 9.6 ( ± 4.8) | — | NA |
| Unemployed or disabled (*n*, %) | 17 (68%) | — | NA |
| Hospitalization/patient | 2.9 ( ± 1.2) | — | NA |
| Clinical dimensions (PANSS scale) |  |  |  |
| PANSS total score | 73.7 ( ± 20.5) | — | NA |
| PANSS positive factor | 11 ( ± 4.1) | — | NA |
| PANSS negative factor | 17.6 ( ± 7.5) | — | NA |
| PANSS disorganized factor | 9.2 ( ± 3.4) | — | NA |
| PANSS excited factor | 5.7 ( ± 1.9) | — | NA |
| PANSS depressed factor | 7.2 ( ± 2.8) | — | NA |
| Antipsychotic equivalence dose (OLZ-Eq, in mg) | 19.6 ( ± 6.6) | — | NA |

CTL, control subjects; NA, not applicable; m/f, male or female; OLZ-Eq, antipsychotic dosage using Olanzapine equivalency; PANSS, positive and negative syndrome scale; PDI: Peters *et al.* Delusions Inventory 21 items scale; r/l: right or left; SCZ, patients with schizophrenia; y.o., years old.

the two groups based on the type of sensory evidence. In trials with unambiguous sensory evidence, the effect of prior congruency remained significant in patients with SCZ ($F_{1,125} = 9.11$, $P = 0.003$), in contrast with CTLs (interaction group × prior congruency: $F_{1,248} = 7$, $P = 0.01$). However, in trials with ambiguous sensory evidence, the combination of prior and sensory evidence was similar between the two groups (interaction group × sensory evidence × prior congruency $F < 1$, $P = 0.8$). Similar to the CTLs, the effect of sensory evidence on patients with SCZ was larger in trials with congruent priors than in those with incongruent priors (interaction prior congruency × sensory evidence: $F_{1,334} = 4$, $P = 0.05$). Together, these results reveal that patients with SCZ integrated prior and sensory evidence differently from the CTLs, and they were incapable of disregarding prior evidence when sensory evidence was unambiguous.

Excluding the potential confounding effect of speed/accuracy trade-offs, similar reaction time (RT) patterns were observed for both groups. Both groups responded faster in cases of sensory evidence (Fig. 1b, right panel; $F_{1,1,127} = 305$, $P < 0.001$) and prior congruency (Fig. 1c, right panel; $F_{1,665} = 13$, $P < 0.001$). The patients with SCZ responded more slowly than the CTLs (Fig. 1b, right panel; $F_{1,973} = 24$, $P < 0.001$). No interaction was significant ($P > 0.075$), although the patterns of variation for RT closely resemble those for the confidence rate, as was evident after comparing the two columns of Fig. 1b,c.

**Model predictions**. To compare the participants' behaviour with predictions from exact or circular Bayesian inference, we constructed different models, the predictions of which are displayed in Fig. 2. Note that all of the predictions and data are now reported as logits and not as probabilities. Thus, the chance logit $L_c = \log(\frac{c}{1-c})$ is predicted as a function of the likelihood logit $L_s = \log(\frac{P_R}{1-P_R})$, with $P_R$ and $1 - P_R$ corresponding to the colour proportion of the fishes in the right and left lake,

respectively. The prior logit was defined as the ratio of the sizes of the two baskets, $L_p = \log(\frac{S_r}{S_L})$, with $S_r$ and $S_L$ corresponding to the size of the right and left fish baskets, respectively. For example, if the right basket was twice as large as the left basket, then the prior logit was log(2). Note that logits can vary between negative infinity (when certain left lake) and positive infinity (when certain right lake). Only trials with ambiguous sensory evidence (that is, finite likelihood logits) were used to fit the model parameters. The model prediction for unambiguous trials is reported in Supplementary Fig. 2. To avoid numerical issues, the confidence $c$ was re-scaled to range between 0.01 and 0.99 (these boundaries were chosen to not significantly affect the results reported here). To avoid confusion, note that the term 'prior' is not used to describe the probability of the sample given for each lake as it has been in previous studies. We interpreted the (sample + fish) proportions as sensory evidence because all of the corresponding information was visually available at the time of the response. In contrast, the basket size information was provided before the response display, and the response required a prediction based on memorized information. Also note that the target fish colour (for example, red fish) did not vary across trials. Thus, the prior was not predictive of the visual display.

The simple Bayes model (Fig. 2a) assumed that sensory evidence and prior information are directly combined using Bayes' theorem. Thus, the likelihood logit and the prior logit were summed. The weighted Bayes model allowed the participants to assign different degrees of trust to the sensory information (that is, fish proportions) and prior (that is, basket size) information. Thus, this model had two free parameters, the sensory weight, $w_s$, and the prior weight, $w_p$ (Fig. 2b). For example, the participants might have believed that even if the baskets had indicated a left choice, a non-zero probability $(1 - w_p)$ would still exist stating that this prediction was incorrect (due to sensory noise, imperfect memory, the fisher changing his mind, a trick of the experimenters and so on). The weighted

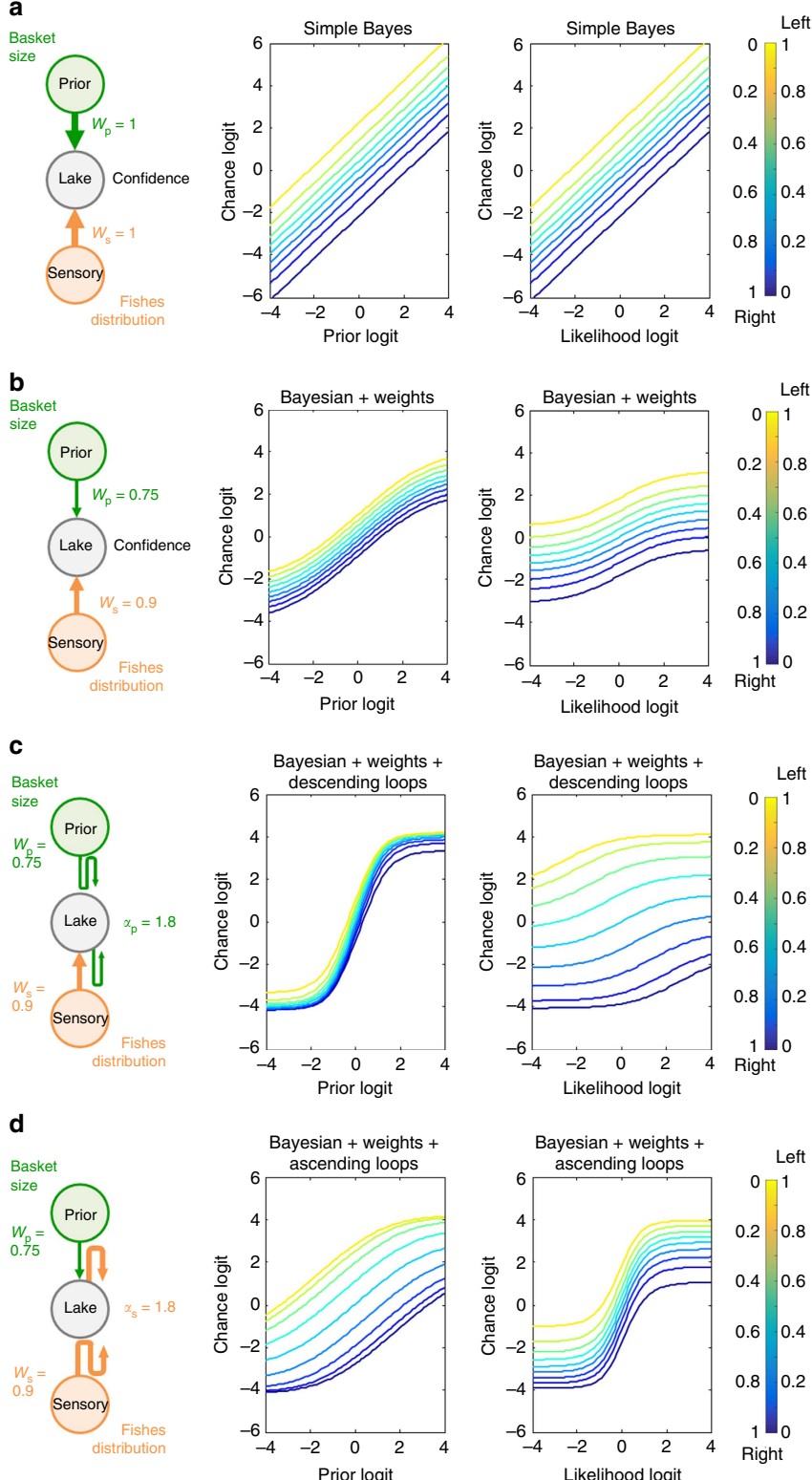

**Figure 2 | Model predictions.** The predictions of each model are plotted as the chance logit against the prior logit and against the likelihood logit (middle and right panels, respectively). In the plots, the likelihood and prior information is colour-coded (from dark blue to yellow) according to the true probability that the fish originated from one of the two lakes in the Fisher task. The simple Bayes model is presented in **a**, with the strength of the prior and the value of the likelihood equal to 1 ($w_p = 1$; $w_s = 1$). The first variant of the model (that is, the weighted model) is presented in **b**, in which the strength of the messages was manipulated ($w_p = 0.75$; $w_s = 0.9$, arrow size in the graph model). This model enables the modulation of the chance/prior/likelihood relationship. The circular inference model is implemented in **c,d**. The addition of inhibitory factors at each level of the hierarchy enables the flow of information to be controlled. This inhibitory control can act independently for feed-forward and feedback propagation. Predictions for large levels of descending loops ($\alpha_p = 1.8$) are presented in **c** whereas predictions for large levels of ascending loops ($\alpha_s = 1.8$) are presented in **d**.

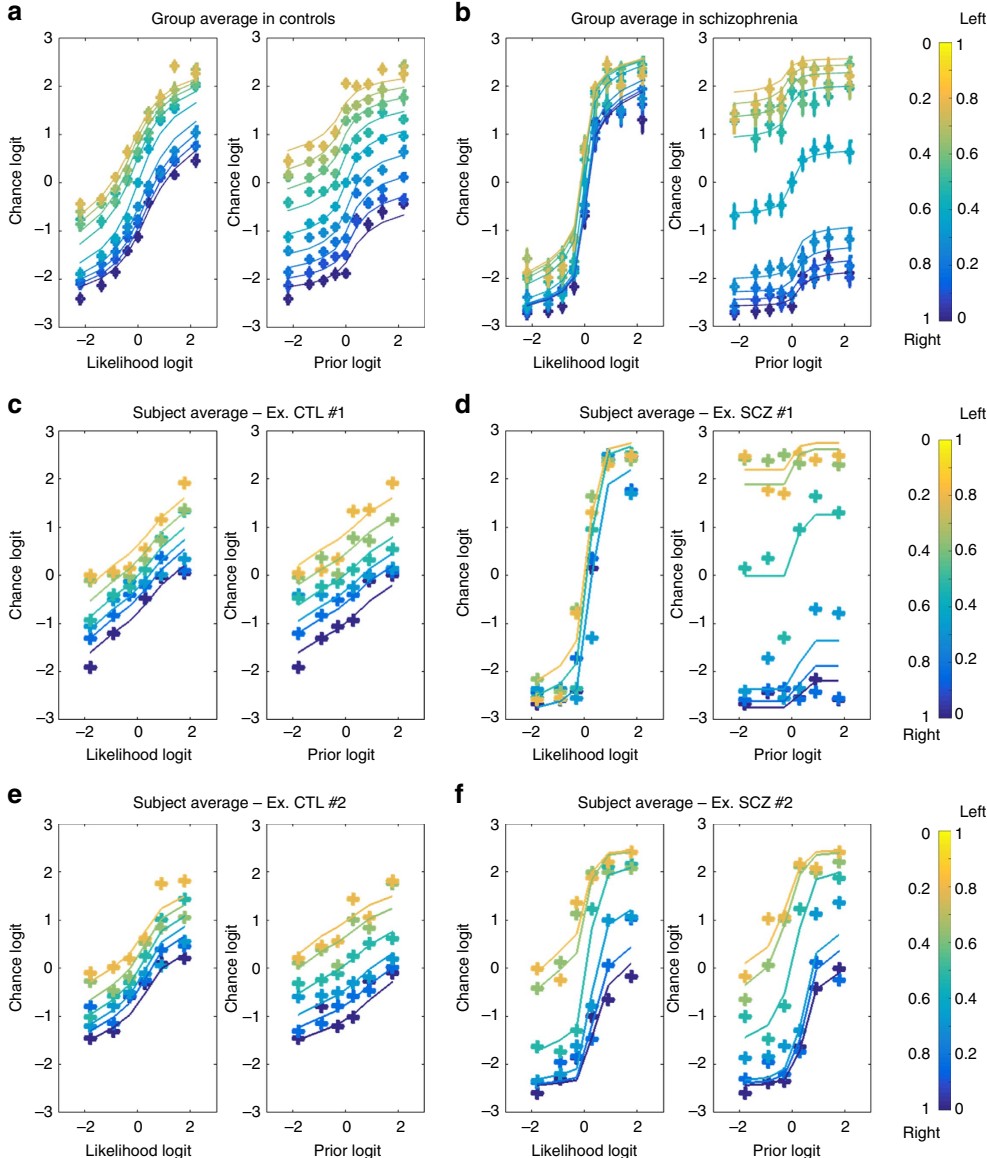

**Figure 3 | Participant confidence and predictions based on the circular inference model.** The mean chance logit is plotted as a function of the likelihood logit or the prior logit. The likelihood and prior information is colour-coded (from dark blue to yellow) according to the true probability that the fish originated from one of the two lakes in the Fisher task. For each plot, the data fits obtained using the circular inference model are overlaid as solid lines using the same colour code. (**a**) Chance logit plotted against the likelihood (left) and prior (right) among healthy CTLs; (**b**) Chance logit plotted against the likelihood (left) and prior (right) among the patients with SCZ. (**c,e**) Data from two CTL participants. Note that one participant (**e**) exhibited a tendency to jump to conclusions (sigmoidal), whereas the other (**c**) made Bayesian decisions (linear) in the Fisher task. (**d,f**) Data from two participants in the SCZ group. Participant (**d**) exhibited an extreme jumping-to-conclusions pattern compared with participant (**f**); once again, however, the circular inference model provided the best fit for these data despite their heterogeneity. (Data and model fits for each participant are provided in Supplementary Figs 3 and 4.)

Bayes model is the binary-choice equivalent of the weighted combination of sensory and prior estimates used in many other studies for continuous variables.

Finally, by including two additional free parameters, $\alpha_s$ and $\alpha_p$, which represented the strength of ascending loops (sensory over-counting) and descending loops (prior over-counting), respectively, the circular inference model enabled the sensory evidence, the prior or both to be 'reverberated' and counted multiple times (Fig. 2c; see 'Methods' section for a detailed description of the implementation and optimization of these models). The specific case of a no-reverberation model has also been tested and is presented in the Supplementary Material.

**Comparison between predictions and participants' behaviours.** The chance logits ($L_c$) for the CTLs and patients with SCZ as a function of likelihood logit ($L_s$) and prior logit ($L_p$) are provided in Fig. 3a,b, respectively. The behaviours of the participants (in the SCZ group and, to a lesser extent, the CTL group) significantly differed from the simple Bayes prediction shown in Fig. 2a. In particular, Bayes' rule predicted linear curves for slope 1 for all of the curves. In contrast, the experimental curves were sigmoidal.

The weighted Bayes model predicted the saturation of the curves around the extreme prior and likelihood values (see Fig. 2b). However, the weighted Bayes model also predicted that the slopes of the curves approximately

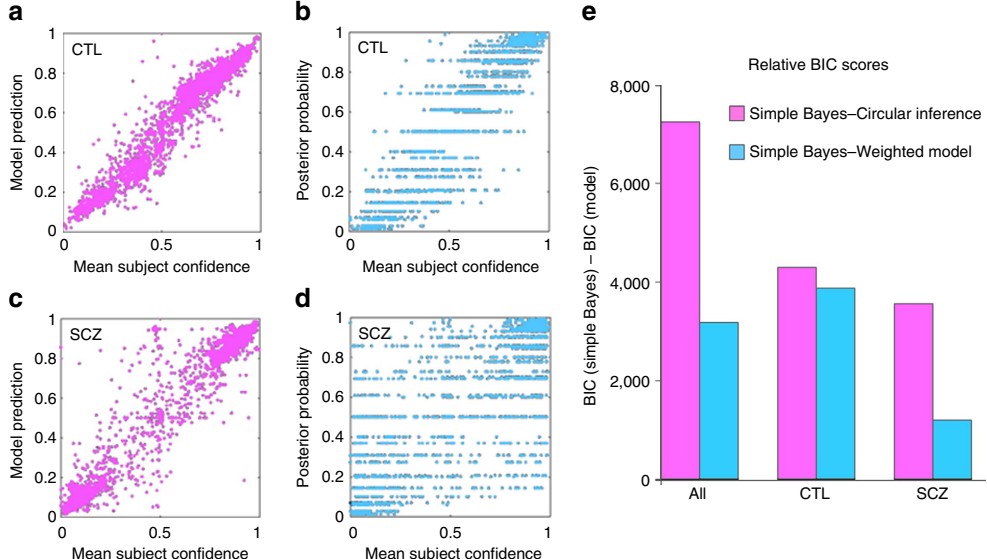

**Figure 4 | Quality of the model fits.** The predictions of the circular inference model (including the weights and loops, shown in purple) are presented for the CTLs (**a**) and for the patients with SCZ (**c**); the predictions of the simple Bayes model are presented for the CTLs (**b**) and the patients with SCZ (**d**; shown in sky blue). The relative gain of the implemented optimisation steps (that is, weights and loops) was determined using relative BIC scores (**e**). The BIC score of the circular inference model (in purple) and the reduced weighted Bayes model (without loops, in blue) was compared with that of the simple Bayes model. The circular inference model optimally captured the heterogeneity of the SCZ sample (**c,e**).

0 should not exceed 1 and that all of the curves should be parallel to each other. These predictions were clearly incorrect, especially regarding the prior logit and the likelihood logit associated with patients with SCZ. Slopes larger than 1 suggest that the same evidence is accounted for more than once. The fact that the slope of $L_s$ is larger when $L_p$ approximates 0 (and vice versa) indicates that prior and likelihood are not treated as two independent sources of information. Rather, the likelihood information is corrupted by the prior information, or vice-versa, before they are combined. Both phenomena can be considered as a signature of circular inference (see 'Methods' section and Fig. 2c).

The weighted bias and circular inference models were individually fitted on each participant's responses (see 'Methods' section). To estimate the models' performances while accounting for the difference in their number of free parameters (four per participants for the circular inference versus two for the weighted Bayes versus none for the simple Bayes), we measured each model's Bayesian Information Criteria (BIC) score (Fig. 4). Smaller BIC values denote a better fit. Over all the participants, the BIC of the circular inference model (1,220) was considerably smaller than that of the simple Bayes model (8,464) and the weighted Bayes model (5,293), confirming that the circular inference model provided a better explanation of the data than the two other models. The same was true within both the CTL and SCZ groups. Interestingly, the performance of the circular inference model surpassed that of the weighted Bayes model (BIC = − 4,730 for circular inference versus BIC = − 4,313 for weighted Bayes) even for the CTLs, suggesting that the CTLs also displayed some degree of circular inference. To confirm these results, we created a Bayesian model comparison for group studies[31] and found a strong dominance of the circular inference model over the simple and weighted Bayes models (CTL group: $\alpha = 25.99$ for circular inference versus $\alpha = 1$ for both the weighted and simple Bayes models; patient group: $\alpha = 24.01$ for circular inference versus $\alpha = 2$ for both the weighted and simple Bayes models, respectively).

Subsequently, all of the reported parameter values (that is, the parameters for weights and loops) refer to those fit with the circular inference model. In Fig. 1b,c as well as Fig. 3a,b, the average circular inference model prediction is presented as an overlaid solid line for the CTL and SCZ groups. Example sessions averaged at the participant level are displayed in Fig. 3c–f (Individual data for the whole sample and model fits are provided in the Supplementary Material). Note that the optimized circular inference model accounted for the behaviour of the SCZ and CTL groups with considerable accuracy (Fig. 4a–d). In particular, it successfully captured the inter-participant variability of the SCZ group (Supplementary Figs 3 and 4).

**Parameter estimates of the circular inference models**. The mean parameter values for the two groups are shown in Fig. 5a. The mean sensory weight ($w_s$) and prior weight ($w_p$) of the CTLs were 0.64 and 0.56, respectively. This result is not surprising given the qualitative nature of the basket size information (see 'Methods' section), while the smaller sensory weights might reflect uncertainty about the exact fish proportion in the two lakes. The amount of ascending loops ($\alpha_s$) and descending loops ($\alpha_p$) was moderate in the CTLs, although the contribution of these loops to behaviour remained significant (see the BIC scores). Overall, the behaviour of the CTLs was relatively close to Bayesian, which accounts for the acceptable fit between their responses and the simple Bayes model (Fig. 4b). In contrast, the simple Bayes model failed to capture the behaviour of the SCZ group (Fig. 4c).

The SCZ group differed from the CTLs as demonstrated by the significant effect of parameter type ($F_{3,144} = 134.3$, $P < 0.000$). The interaction between parameter type and group was also significant ($F_{3,144} = 18.1$, $P = 0.001$). In particular, the patients with SCZ severely over-counted the sensory evidence, as demonstrated by the amount of ascending loops, $\alpha_s$, which was significantly larger than that for CTLs ($z = − 5.25$, $P < 0.001$). The mean sensory weight, $w_s$, was also slightly larger in patients with SCZ than CTLs ($z = − 3.05$, $P = 0.046$).

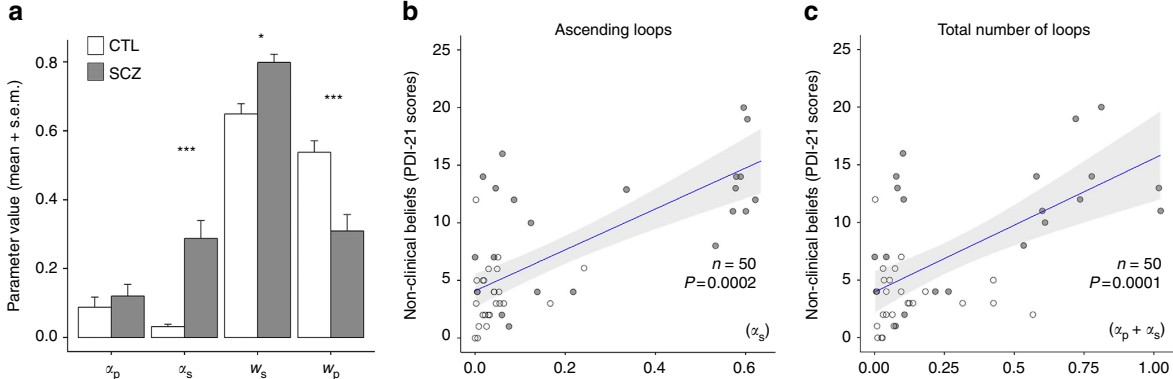

**Figure 5 | Group comparisons of model parameters. (a)** The strength and number of ascending loops in the hierarchical model in the patients with SCZ ($n = 25$, grey) were significantly increased compared with those in the matched healthy CTLs ($n = 25$, white). The data are presented as the mean $\pm$ s.e.m. **(b)** Non-clinical beliefs (PDI-21 scores) and the strength of the ascending loops were positively associated in the CTL and SCZ groups, regardless of WM performance (regression analysis with backward selection). **(c)** Similarly, non-clinical beliefs were significantly associated with the total number of loops in the whole sample. Regression lines are plotted with their 95% confidence intervals.

In contrast, the mean prior weight, $w_p$, was substantially smaller in the patients with SCZ than in the CTLs ($z = 4.67$, $P < 0.001$; see Fig. 1c,d); however, the two groups did not differ with regard to the extent to which the priors were over-counted, as evaluated by the amount of descending loops, $\alpha_p$ ($P = 0.99$).

Thus far, we applied the model to ambiguous trials with finite likelihood logits (that is, trials in which probabilistic reasoning was required). In unambiguous trials, the participants were instructed to click on the extremity of the scale, which the CTLs did (Fig. 1c). Interestingly, patients with SCZ continued to be influenced by the prior in those trials (Fig. 1d). The particular case of unambiguous trials is presented and discussed in the Supplementary Material.

**Prior weight was correlated with working memory performance.** The parameter $w_p$ must be treated with some caution because the prior is somewhat qualitative, in contrast with the sensory evidence. For example, part of the logit non-linearity might be because of the distortion in perceived sizes. However, because we have no particular reason to believe that this effect would differ between the CTL and SCZ groups, we continued to interpret differences in $w_p$ as representing differences in prior weighting. Because this information must be memorized, the between-group differences in $w_p$ might solely be a manifestation of the types of cognitive dysfunction that are highly prevalent in patients with SCZ[32,33]. We collected independent measures of working memory (WM) performance through post-experimental psychometric tests. The WM performance of the patients with SCZ was poorer than that of the CTLs ($P = 3.1e–06$, Table 1) and correlated with $w_p$ over the whole sample ($r = 0.29$, $P = 0.039$). However, WM did not differentially affect the other parameters values within the SCZ and CTL groups (for example, non-significant correlations were observed with regard to $w_s$, $\alpha_s$ and $\alpha_p$, but a positive interaction was found between WM and the four model parameters, $P = 0.02$).

**Ascending loops were correlated with positive SCZ symptoms.** The amount of ascending loops ($\alpha_s$) were positively correlated with non-clinical beliefs, which were measured across the whole sample using the 21-item Peter's Delusional Inventory (PDI-21) scale ($b = 0.59$, $F_1 = 15.6$, $P < 0.000$ and $b = 1.1$, $F_1 = 16.7$, $P < 0.000$; Fig. 5b,c). However, this correlation was not significant among the CTLs. A regression model using

backward selection excluded the potential association between beliefs and WM performance ($P = 0.3$).

Additional analyses were restricted to the SCZ group. The associations between the model parameters and psychopathology were tested using the Positive and Negative Syndrome Scale (PANSS) scores measured in the SCZ sample. Global severity was associated with the number of ascending loops (or sensory over-counting), $\alpha_s$, ($r = 0.48$, $P = 0.03$) and the total number of loops, $\alpha_s + \alpha_p$ ($r = 0.54$, $P = 0.01$; Fig. 5b). More precisely, ascending loops were associated with the severity of psychotic symptoms (PANSSp; $r = 0.45$, $P = 0.05$; Fig. 6a) but not with the severity of negative symptoms ($P = 0.14$), with a significant interaction ($P = 0.04$). Interestingly, $\alpha_s$ was particularly strongly associated with delusions (item P1, $r = 0.61$, $P < 0.01$), hallucinations (item P3, $r = 0.47$, $P = 0.03$), and unusual thought content (item G9, $r = 0.45$, $P = 0.05$).

**Descending loops were correlated with negative SCZ symptoms.** Although the severity of descending loops, $\alpha_p$, did not significantly differ between the SCZ and CTL groups, it varied greatly from patient to patient. Among patients with SCZ, $\alpha_p$ (but not $\alpha_s$) was correlated with the severity of negative symptoms (PANSSn; $r = 0.51$, $P = 0.02$), with a significant interaction ($P = 0.03$). The $\alpha_p$ parameter was particularly correlated with emotional withdrawal (item N2, $r = 0.5$, $P = 0.02$), poor rapport (item N3, $r = 0.62$, $P = <0.01$), and lack of spontaneity (item N6, $r = 0.49$, $P = 0.03$). Thus, the severity of negative SCZ symptoms might be associated with an over-counting of prior beliefs.

**Both loops were correlated with disorganized symptoms.** In addition, disorganized symptoms (PANSSdis) were correlated with $\alpha_s$ ($r = 0.46$, $P = 0.04$) and $\alpha_p$ ($r = 0.48$, $P = 0.03$) as well as with $\alpha_s + \alpha_p$ ($r = 0.54$, $P = 0.01$; Fig. 6c). The total number of loops ($\alpha_s + \alpha_p$) was particularly strongly associated with conceptual disorganization (item P2, $r = 0.55$, $P < 0.00$), difficulty in abstraction (item N5, $r = 0.51$, $P = 0.02$) and poor attention (item G11, $r = 0.45$, $P = 0.05$).

In contrast, affective symptoms were not significantly correlated with the parameters evaluated using the PANSS excitation and depressive factors. This result suggests that different mechanisms are involved in these dimensions, which are common to both SCZ and bipolar disorder.

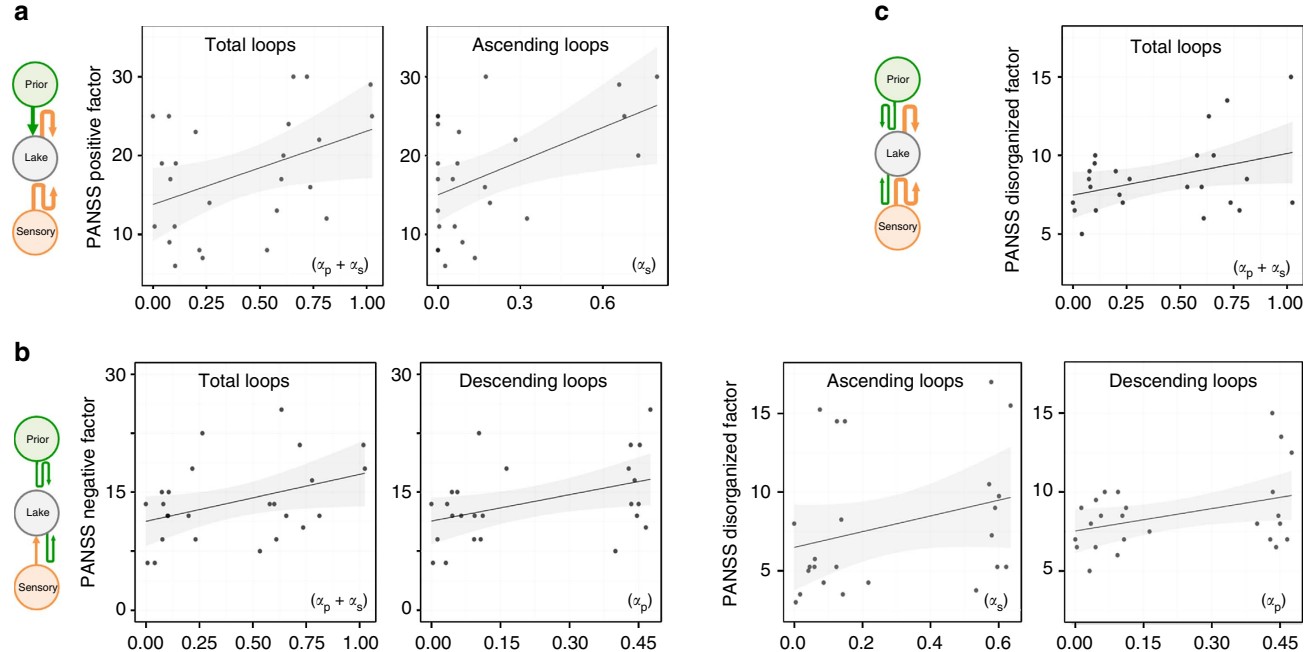

**Figure 6 | Correlation analyses between the parameters and the clinical dimensions of SCZ.** (**a**) Positive SCZ symptoms, which were measured using the PANSS positive subscale, were correlated with the number of ascending loops ($\alpha_s$, $r = 0.46$, $P = 0.04$) and the total number of loops (loops, $r = 0.45$, $P = 0.05$). These results correspond to the over-counting of the sensory evidence (orange arrows in the graph model); (**b**) Negative symptoms, which were measured using the PANSS negative subscale, were correlated with the number of descending loops ($r = 0.51$, $P = 0.02$) and the total number of loops ($r = 0.46$, $P = 0.04$), corresponding to the over-counting of the prior information (green arrows). (**c**) Disorganized symptoms, which were measured using the PANSS disorganized subscale, were correlated with both ascending ($r = 0.48$, $P = 0.03$) and descending ($r = 0.46$, $P = 0.04$) loops, which indicates the global over-counting of the feed-forward and feedback information. Regression lines are plotted with their 95% confidence intervals.

## Discussion

Using the Fisher task paradigm, we compared the degree of confidence that the SCZ and CTL groups derived from combining prior information and sensory evidence. By fitting a model inspired by a hierarchical neural network with impaired E/I balance, we interpreted the between-group differences in terms of specific impairment in Bayesian inference mechanisms. The model successfully captured not only the group-averaged behaviour but also the large inter-participant variability, within both the SCZ and CTL groups.

The most salient results of this study are the large over-counting of sensory evidence and the under-weighting of priors in SCZ. Importantly, the over-counting of sensory evidence cannot be simply explained by an overly large amount of trust placed in the sensory evidence compared with the prior. Although the patients assigned slightly more trust (that is, sensory weight, $w_s$) to the likelihood information, most of the difference between the patients with SCZ and the CTLs came from the large number of ascending loops, $\alpha_s$, in the patients with SCZ. In particular, when the fish proportion only slightly differed between lakes, patients with SCZ showed strong differences in confidence (that is, a tiny change in the likelihood logit near 0 might cause a large change in the chance logit). This result cannot be explained by Bayes' theorem alone, regardless of how reliable the sensory information was assumed to be. Even when maximal trust was placed in the sensory evidence ($w_s = 1$), a change in the likelihood logit caused an equal change in the chance logit.

We attributed this phenomenon to the presence of ascending inference loops that allow sensory evidence to be combined with itself and the prior multiple times[21]. More concretely, a slightly higher proportion of fishes in the right lake might have generated a top-down prediction favouring the right lake. In the

absence of a proper inhibitory control, this 'redundant' prediction (as opposed to an independent source of prior information such as basket size) could be re-combined with the sensory and prior data, further inflating the patients' confidence in the right lake interpretation. This phenomenon might have caused the patients with SCZ to experience high confidence levels that were disproportionate with the available sensory data.

Crucially, the amount of ascending loops was correlated with the amount of non-clinical and delusional beliefs. These findings are compatible with the theoretically predicted effects of circular inference, which include the inappropriate causal attributions that might be the root of psychotic symptoms[21,34]. This result also suggests that psychotic symptoms are linked to sensory over-counting and not a dominance of prior beliefs.

The sensory over-counting interpretation of SCZ might explain the paradoxical finding that patients with SCZ, most notably those with psychotic symptoms, can be less susceptible to visual illusions[35–37]. This attenuation has also been observed in people who exhibit high delusional belief scores[38], but it appears to be independent of affective symptomatology[39] in accordance with the present findings. Within the framework of Bayes' theory, illusions likely arise from weak or conflicting sensory evidence and from strong prior evidence[40]. Interestingly, this pattern of results regarding the illusions associated with SCZ, although frequently reported, has not always been replicated in the literature[9]. We predict that patients with preeminent negative symptoms (in whom we found an increased amount of descending loops) will exhibit unchanged, or even augmented, sensitivities to illusions.

Our results are also consistent with previous studies describing jumping to conclusions in people with SCZ (for example, refs 28,41,42) and with reports showing that people with higher

PDI-21 scores tend to request fewer beads before making a decision in probabilistic reasoning tasks[26]. More discussion regarding this literature is available in the Supplementary Material.

Finally, comparing our approach with other models in which the prior and sensory estimates were summed with weights corresponding to their assumed precision is interesting. This comparison includes both free energy models[43] and predictive coding models[44]. Note that such models are appropriate for continuous variables, such as position or direction, but not for binary choices, such as right or left. For example, this finding would lead to an aberrant confidence estimate of '2' when both the prior and likelihood unambiguously favour the right lake (that is, an estimate of 1). For binary variables, the closest equivalent is the weighted Bayes model (that is, a Bayesian model in which the precision of the sensory and prior information is allowed to vary). If the variables were continuous and the probability distributions were Gaussians, then circular inference might also result in a weighted combination of prior and sensory estimates. However, in the presence of strong circular inference, the weights reflect the amount of loops $\alpha_s$ and $\alpha_p$ and not with the subjective precision of sensory and prior information. Please note that our model stays compatible with alternative computational frameworks. Jumping from attractor to attractor in a dynamic framework could for instance be interpreted as the generation of aberrant beliefs in the circular inference framework.

The manner in which cognitive impairments might have interfered with our results was of particular concern because altered WM might affect the jumping-to-conclusions bias[45]. Although associations between executive function and positive symptoms have emerged from meta-analyses[46], these associations have not always replicated[47], most notably because of the lack of WM manipulation during data collection. By changing the prior information in each trial, we were able to reveal only the influence of WM on prior weight ($w_p$). One explanation for the tendency of patients with SCZ to mistrust the basket information is that they could not always remember it well. In fact, WM in patients with SCZ was correlated with prior weight, $w_p$, but not with the parameters directly associated with the jumping-to-conclusions bias (that is, $\alpha_s$, $w_s$ or even $\alpha_p$). This result suggests that WM impairments are not a major factor in the over-counting of sensory information and its association with psychosis, which is consistent with the data showing the independence of cognitive deficits and psychotic symptoms in people with SCZ[48,49].

To date, the central challenge of SCZ research has been the lack of an integrative model that accounts for the heterogeneity of the disorder. By fitting a circular inference model, we were able to capture the three main clinical dimensions of SCZ: positive symptoms (that is, psychosis), negative symptoms and disorganization. These different clinical dimensions rely on partially overlapping impairments, which correspond to the cortical circuit correlates of the components of the circular inference model[21]. This holistic approach to SCZ constitutes the major strength of the circular Bayesian framework.

In addition to the association between ascending loops and psychosis, we found a correlation between negative symptoms and descending loops, which suggests that the tendency of prior beliefs to overcome the sensory evidence is one of the factors that lead to negative symptoms. We also found a correlation between all of the loops and the dissociative features of SCZ. This observation is in line with our theoretical predictions because the presence of both types of circular inference results in dissociations between low-level sensory representations and their high-level interpretations[21]. This last result highlights the need to question and reconsider disorganized symptoms as the ultimate stage of the disorder. This perspective is reminiscent of the seminal view of Eugen Bleuler, who proposed that the 'splitting of psychic functions' was a core feature of SCZ more than a century ago[50].

Finally, one intriguing result of this study was that circular inference was necessary to account for the responses of both patients with SCZ and CTLs. Only a minority of CTLs could be considered as ideal Bayesian observers (that is, those who matched a linear fit). Although we did not find a correlation between the PDI score and ascending loops in our CTLs, such a relationship might emerge in a larger, more heterogeneous population (that is, in terms of range of beliefs), indicating that moderate circular inference is not necessarily pathological (see also refs 43,51). This finding appears compatible with recent physiological recordings coupled with optogenetic inhibition, which revealed that reverberation, with notably recurrent inputs to sensory areas, is essential for accurate perception[52]. Future studies will explore circular inference and its implications for perception and decision making in CTLs.

## Methods

**Sample**. A statistically valid sample size was defined at 21 participants/group, based on an *a priori* power calculation performed on confidence rates from pilot test data. Considering a pessimistic 15-to-20% drop-out rate, we enrolled 25 participants in each group (that is, 25 patients who satisfied the Diagnostic and Statistical Manual, Fourth Edition, Text Revision (DSM-IV-TR) criteria for SCZ[53] and 25 healthy CTLs who were matched for age, gender and years of education (see Table 1). Because no technical problems were encountered, all 50 participants were included, and their data were analysed. All of the participants were adults with normal or corrected-to-normal vision. They were all tested for spatial attention using the Bells cancellation task, cognitive inhibition using the Stroop interference task, WM using the digit span task and non-clinical beliefs using the PDI-21 (ref. 54). Psychopathology was specifically explored in the patient group using the PANSS[55]. A senior psychiatrist confirmed the absence of psychiatric symptoms among the CTLs using the Mini International Neuropsychiatric Interview (MINI-DSM-IV)[56]. The exclusion criteria were the presence of an Axis-II diagnosis, a secondary Axis-I diagnosis, a neurological or sensory disorder, a history of drug abuse based on the clinical interview and at-admission urine tests and an IQ below 80. CPP Nord-Ouest IV provided ethical approval for this study. All the volunteers provided written informed consent.

**Stimuli and design**. The participants viewed stimuli on a LCD monitor placed 50 cm away. After successfully performing three separate training blocks with the prior only, likelihood only and the combination of the prior and likelihood information, all 50 participants completed the Fisher task, which consisted of 8 blocks of 30 trials each for a total of 240 trials/participant. The participants were allowed to take breaks between each block. Figure 1a shows the event sequence for one trial in the task.

After a fixation cross was presented for 800 ms, the prior, which consisted of two fish baskets (left and right), was presented for 1,000 ms. The participants were told to interpret the size of these baskets as the degree of preference that the fisher had for the left or right lake. After a 50-ms delay during which the fixation cross was again presented, the likelihoods were presented at the same locations where the baskets were presented. The likelihood consisted of the proportion of black and red fishes found within the left and right lakes. A fixed number of fishes was included in each lake, and the values for the prior and the sensory evidence associated with the two lakes were symmetrical (for example, ratios of 2:8 and 8:2 for the left and right, respectively).

The participants were instructed to report their confidence that a red fish originated from one of the two lakes. A red fish was used for all trials. The participants replied using a semi-circular scale that ranged from 100% confidence that the fish had been caught in the left lake to 100% confidence that the fish had been caught in the right lake. To avoid misinterpretation, we instructed participants to answer at the extremity of the scale (that is, 'totally certain' for unambiguous trials). RTs were recorded. After the response was collected, the fixation cross was presented alone during a 200-ms delay before the onset of the next trial. The prior and likelihood information were manipulated for each trial and could take the following values: 0.1, 0.2, 0.3, 0.4, 0.5, 0.6, 0.7, 0.8 and 0.9 for the prior and 0, 0.1, 0.2, 0.3, 0.4, 0.5, 0.6, 0.7, 0.8, 0.9 and 1 for the likelihood. The trial sequence for each block was generated using a pseudo-random algorithm.

**Predictions based on simple Bayesian inference**. Three distinct variables were manipulated in the Fisher task. First, the size of the basket reflected the prior probability for each lake. For example, a basket size ratio of 8:2 corresponded to a prior probability of 0.8 for the right lake, whereas a 4:6 ratio represented

a prior of 0.4. Note that the prior was intentionally qualitative in nature. Thus, although we were unable to compare different participants, each participant could have a different interpretation of the mapping between basket size and prior information (see circular inference model).

Second, the proportion of fishes in each lake reflected the sensory evidence in the form of the likelihood that the fish originated from the right or left lake. In contrast to the prior, the likelihoods could be precisely manipulated by changing the proportion of black and red fishes in each lake. For example, if 40 black fishes and 10 red fishes were present in the left lake, and 10 black fishes and 40 red fishes were present in the right lake, then the likelihood of a red fish, $P_R$, was 0.80 for the right lake.

Finally, the participant's response, $c$, was the normalized position on the confidence scale (Fig. 1a). To avoid numerical issues, $c$ was rescaled to range between 0.00001 and 0.99999. The chance logit was defined as $L_c = \log\left(\frac{c}{1-c}\right)$. Note that $L_c$ is positive when the participant is confident that the fish comes from the right lake, negative when the participant is confident that the fish comes from the left lake, and 0 if the participant is completely uncertain. $L_c$ is reported as a function of the likelihood logit, $L_s = \log\left(\frac{P_R}{1-P_R}\right)$, with $P_R$ representing the proportion of red fish in the right lake and the log prior logit, $L_p = \log\left(\frac{S_R}{S_L}\right)$, where $S_R$ and $S_L$ correspond to the size of the right and left baskets, respectively.

An optimal Bayesian observer should apply Bayes' theorem to compute the posterior probability of the two lakes as the product of the likelihood and the prior (normalized to a sum to 1). We called this hypothesis the 'simple Bayes' model, which can be expressed in terms of log odds ratios, $L_c = L_s + L_p$. The Fisher task predictions of the 'simple Bayes' model are provided in Fig. 2a.

**Weighted Bayes model.** To include the possibility that the participants might consider the sensory cues (that is, fish colour) and prior information (that is, basket size) 100% reliable, we interpreted the hierarchical structure of the Fisher task using a small normative causal model composed of three binary variables (see Fig. 2, left panels). The fisher's preference (top variable, $x_p$) imperfectly predicts the origin of the fish (middle variable, $x_c$). In turn, fish origin imperfectly predicts the sensory evidence provided by the fish colour and the proportions (bottom variable, $x_s$). For the sake of simplicity, we assumed that all three variables could take only one of the two possible states: right lake or left lake (see Fig. 1a). However, note that a more detailed normative model (for example, one that included uncertainty about the real proportion of fishes in each lake and the relative size of the two baskets) would make similar predictions.

The probability of lake choice as a function of fisher preference, $p(x_c|x_p)$, is parameterized by a prior weight, $w_p$, such that $w_p = p(x_c = R|x_p = R) = p(x_c = L|x_p = L)$; $w_p$ denotes the reliability or the trust placed by the participant in the basket size information. Similarly, $p(x_s|x_c)$ is parameterized by a sensory weight, $w_s$, such that $w_s = p(x_s = R|x_c = R) = p(x_s = L|x_c = L)$ and represents the reliability or the trust placed by the participant in the fish proportion information. In this context, exact probabilistic inference would predict that

$$L_c = F(L_s, w_s) + F(L_p, w_p),$$

where $F(L, w) = \log\left(\frac{we_L + 1 - w}{(1-w)e_L + w}\right)$.

In particular, if the participant trusted both the sensory information and the prior completely ($w_p = w_s = 1$), then this model would be mathematically equivalent to the simple Bayes model.

**Circular inference model.** Rather than being taken into account only once and then added, as in the weighted Bayes model, the sensory evidence and prior can reverberate in the circular inference model because of the E/I imbalance. Rather than building a hierarchical neural network model that would include too many free parameters, we captured the essence of this phenomenon by including only two additional free parameters: $\alpha_s$ represents the number of times the sensory evidence is taken into account redundantly, whereas $\alpha_p$ represents the number of times the prior information is taken into account redundantly.

Each time the sensory evidence is reverberated into a recurrent network, it can be wrongly considered as new additional evidence. At each reverberation, the likelihood is multiplied by itself (that is, the logits are summed) resulting in a redundant contribution, $\alpha_s L_s$. Through the same reasoning, reverberation of the prior information would contribute $\alpha_p L_p$. However, because the participants do not completely trust either source of information, the effective contributions of over-counted information can be summarized as $F(\alpha_p L_p, w_p)$ and $F(\alpha_s L_s, w_s)$. Finally, because these are the results of multiple reverberations up and down a hierarchical circuit, over-counted information will equally corrupt (be added to) the top-down and bottom-up components of inference, rendering the likelihood and prior information completely inseparable. The resulting simplified equation for the circular inference model is

$$L_c = F\big(L_s + F(\alpha_p L_p, w_p) + F(\alpha_s L_s, w_s), w_s\big) + F\big(L_p + F(\alpha_s L_s, w_s) + F(\alpha_p L_p, w_p), w_p\big)$$

Note that this model is equivalent to the weighted Bayes model for $\alpha_s = \alpha_p = 0$. If the weights are also equal to 1, then it is equivalent to the naive Bayes model. In other words, the two previous models are special cases of the circular inference model.

To predict the absolute confidence in Fig. 1b,c, we added a small noise term (with s.d. = 0.02) to the likelihood. Note that the 'absolute confidence' around $c = 0.5$ depends on the variance of the responses, not the mean. Because the noise we added is extremely small, it makes no difference except for patients with SCZ featuring extreme levels of ascending loops. In trials with zero sensory evidence, these patients respond randomly at the extremes of the scale, not in the middle. According to the model, the slope of the logit at approximately 0 is so large that any small noise pushes the response to the right or left of the scale, reproducing this behaviour. The predictions of this 'noiseless' model are shown in Supplementary Fig. 2. The noise was not used in the model fitting procedure or any model results reported elsewhere.

**Optimization of the circular inference and weighted Bayes model.** The circular inference model contained four free parameters per participant ($\alpha_s, \alpha_p, w_s$ and $w_p$), whereas the weighted Bayes model contained two ($w_s$ and $w_p$). These parameters were fit on a participant-by-participant basis by minimizing the mean-squared distance between the predicted and reported chance levels, $c$ (that is, the position of the participant's response on the chance scale), on all of the trials with ambiguous sensory evidence. The mean-squared error appeared appropriate because although the participants were estimating probabilities, their responses were also corrupted by their own motor errors, even when they chose the extremes of the scale. However, we also minimized the KL divergence between $c$ and its prediction. This yielded similar results (not reported here), including those of all statistical tests that reached significance.

**Sample characteristics analysis.** Statistical analyses were conducted using R 3.3. Significance was considered as $P < 0.05$. Continuous variables were compared using paired-samples $t$-tests when normally distributed or using the Wilcoxon rank sum test (i.e., Shapiro–Wilk test $P < 0.05$). Categorical variables were compared using Pearson's $\chi^2$-test.

**Model-free performance analysis.** We analysed the absolute confidence of each group using a general linear mixed model that modelled the parametric effects of sensory evidence, prior congruency and their interaction as fixed effects with repeated measures across participants, whereas the participants were treated as a random factor. Absolute confidence was computed as $|(\text{confidence} - 0.5) \times 2|$. The parametric effect of sensory evidence corresponded to the absolute value of the log likelihood ratio. The parametric effect of prior congruency corresponded to the difference between the log prior for the option with larger log likelihood relative to that for the other option. In this sense, the effect of the relative log prior ratio modelled the strength of the prior belief congruent with the sensory evidence. The group-comparison analyses also modelled the fixed effects of group without repeated measures across participants as well as the two- and three-way interactions with the effects of sensory evidence and prior congruency. We modelled the variance-covariance matrix of the fixed effects with a simple variance component that estimated the contribution of the random effect to the variance of the fixed effects without additional assumptions regarding the contribution to the covariation across the fixed effects. To describe the significant interactions between the parametric effects, we performed breakdown analyses restricted to trials with unambiguous sensory evidence (that is, where the log likelihood ratio = 1) and trials with ambiguous sensory evidence (that is, where the log likelihood ratio < 1). The same set of analyses was conducted on the RTs to reject the hypothesis concerning the confounding effects of a speed-accuracy trade-off between the two groups.

**Exploration of the jumping-to-conclusions bias.** Complementarily, and to facilitate comparisons with previous findings in the literature, the jumping-to-conclusions bias (that is, the tendency to make decisions without enough information), was explored by examining the degree of confidence in the replies. Three different tests were conducted: (1) a group comparison of the mean confidence level $\pm$ the s.d. (Wilcoxon rank sum test); (2) a group comparison of the number of extreme ratings (Wilcoxon rank sum test); and (3) a group comparison of choice quality (that is, whether the participants' choices were in line with the optimal solution according to Bayes' theorem; Pearson's $\chi^2$-test). These analyses are presented in Supplementary Fig. 1.

**Assessing model fit quality.** We used the BIC scores to compare the quality of the model fits for the three implemented models. We approximated the likelihood function of all of the models as normally distributed. The reported BIC score estimates are

$$\text{BIC} = n \log(\sigma^2) + k \log(n),$$

where $n$ is the total number of data points (that is, the total number of trials combined over all participants), $k$ is the number of free parameters (that is, four times the number of participants for the circular inference model, two times the number of participants for the weighted Bayes model and zero for the simple Bayes model), and $\sigma^2$ is the mean-squared difference between the chance estimate $c$ on each trial and the model prediction, averaged over all participants.

**Group comparisons using model parameters.** To assess the different factors that might affect group differences in the model parameters, we computed a multilevel model for mixed designs, with parameter value defined as the dependent variable and parameter type as the within-participants factor (that is, $\alpha_s$, $\alpha_p$, $w_s$ and $w_p$). We also investigated the contribution of WM deficits using a multilevel model with $\alpha_s$, $\alpha_p$, $w_s$, $w_p$ and WM as within-participant factors.

**The association between the parameters and clinical/non-clinical beliefs.** The associations between the parameter values and the severity scores were evaluated using two-tailed Pearson's product moment correlations. The global psychopathology of the SCZ group was measured using the PANSS total score. We referred to the five-factor model of the PANSS to assess the specific effects of the different clinical dimensions observed in SCZ[57] (that is, positive, negative, disorganized, excited and depressed factors). The correlations between these subscores and the different parameters in the models were also examined. Multiple testing corrections were made using the false discovery rate method.

To assess the independent association between non-clinical beliefs (PDI-21 scores) and the participant values of $\alpha_s$, $\alpha_p$ and loops, we included these variables along with WM performance in a regression model using backward selection and replicated the analysis using the 'all subsets' method.

**Code availability.** The Matlab codes implementing the circular inference model are available in the Supplementary Material.

**Data availability.** The data that support the findings of this study are available from the corresponding author upon reasonable request.

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

## Author contributions

All authors conceived the experiment; A.L. and S.D.U. implemented the task and the protocol in psychtoolbox; R.J. enrolled the participants; R.J. and S.D.U. performed the statistical analyses; R.J. and S.D.E. performed the model predictions and computational analyses; and all authors contributed to the interpretation of the findings and to the writing of the manuscript.

## Additional information

**Competing financial interests:** The authors declare no competing financial interests.

