## [Peer Review File · Nature Communications]

Reviewers' Comments:

Reviewer #1 (Remarks to the Author):

Thank you for sending me this paper for review. It presents a novel task, something of a mixture between random dots tasks and jumping to conclusions tasks, which examines how subjects integrate two very different types of information.

The task, the analysis and the results are striking, very innovative and have the potential to have a strong impact on the field. The writing is clear and the paper a pleasure to read.

I do have quite a few comments, and list them below. Although some of these read as very fundamental and hard comments, I would like to emphasize that, first, I think that all of these can be addressed with some additional analyses or extra reporting, and that even if some of them cannot, these are limitations that do not detract from the broader importance of reporting the task, and in some way just attest to the depth, strength and structure of the task, results and modelling.

I should add that I will not agree re-review the paper unless the figures/legends/tables are included inline (including suppl. figures).

major points

- the prior manipulation is poorly explained - please provide more details, and include the exact instructions with translations as supplementary material. The use of a basket suggests that this is how many fish were caught from each of the two lakes, and it is not clear whether the single fish presented was then drawn from these baskets, or from the lakes directly. The methods imply that the baskets were used to indicate the fisher's preference, which might be more understandable.

- To what extent could differences between the groups be due to differences in the understanding of the task instructions? What aspects of the data

speak to this? For instance, if one were to choose subjects with comparable prior weight (which might be one, admittedly imperfect, proxy for the comprehension of the prior manipulation), would the remaining differences still hold?

- Similarly, the scale on which the prior lives is poorly defined, and there may well be a strange function mapping the logits defined by the authors to the perceived logits by subjects. Could such a warp could possibly also provide some of the nonlinearity around zero?

- Patient description is too superficial. The average dose of medication is on the high side with 20mg of olanzapine, suggesting that this is quite a severely ill sample.

- Information about patient characteristics needs to be in the main text
- Suppl. Information about patients needs to contain medication (not just olanzapine equivalents), medication history as far as known, and a characterisation of severity in terms of illness course: years of illness, occupational status and comorbidities, number of hospitalisations etc.

- the notion of 'reverberating loop' is strange, as they're not really loops, but two separate factors, one for the prior and one for the evidence. It would be helpful to clarify this in terms of the language and interpretation. This might be a misunderstanding that could be helped by providing the derivations for the reverberating model.

- Discussion: "in SCZ, tiny changes in the likelihood logit around 0 could cause extremely large changes in the chance logit" This is cryptic for: when the fish proportions differed only slightly, patients nevertheless showed strong differences, I assume? Given that they did not attend much to the prior, it seems that in this difficult problems, something particular does in fact happen. Is there any corroborating evidence about what subjects did e.g. from debriefing? Could this be not an effect of likelihood, but an effect of "difficulty"?

- it is a shame that the reaction times have not been examined. Would the authors not have expected them to correlate with the amount of 'reverberation', maybe $\alpha + \alpha_p$?

- Discussion: "The most salient result of this study is the large over-counting of sensory evidence in schizophrenia" I don't understand this conclusion - the

group effect on the prior is equally large and more significant.

- in the results on correlation with PANSS, there are a number of statements 'correlated with x, but not with y'. These should be supported by direct test using interactions

minor points

- The distinction between prior and evidence is somewhat fuzzy. Could one equally well turn around the two, call 'evidence' the size of the basket, and prior the proportions in the lake. What would the consequences be for the model / interpretation of the results?

- I don't understand the F tests: the values look like they were performed as fixed effects, but the methods state they were mixed effects. Could you please clarify how the degrees of freedom arise?

- p6: I don't understand how the lines in the figures, which seem to show rather striking differences in the effects of congruency between the groups is compatible with "on the remaining trials, prior congruency modulated the effect of sensory evidence similarly in the two groups (three-way interaction: $p=0.80$ ", particularly in the view of the significance of visually far smaller effects, e.g. the impact of prior congruency per se in the SZ group.

- the model assumes that the information extracted from both visual stimuli is the same as that extracted from one. Can this assumption be substantiated?

- please provide a derivation of the recursive model - it is not obvious to me exactly how this is arrived at.

- please superimpose the model fits on the data in figure 1b,c. This would clearly show how good a model this is, and that it can capture ever strikingly complex and counterintuitive aspects of the data.

- parameter estimates, p10: "In the CTLs, the mean sensory weight () was close to 1 (Fig. 5a), but the prior weight () was less than 1." -> this is not compatible with the figure, which shows 0.6 and 0.5, respectively!

- "However, WM did not differentially affect the other parameters values within the SCZ and CTL groups ($p=0.24$)." why is there only one p value for all other

parameters for both groups? How exactly was this tested?

- please label the axes in 5b,c, and 6.

language:

- 'fishes distribution' -> 'fish distribution' or 'distribution of fishes'

- 'severity of ascending loops' -> disorders have a severity, but not loops.

maybe strength?

Reviewer #2 (Remarks to the Author):

The authors compare schizophrenia patients and controls on a decision making task that requires integration of prior information with existing evidence. Computational analysis of the behavior on the task explains the differences between the groups in terms of the strength of ascending and descending fibers, which also differentially correlate with different types of clinical symptoms. The hypothesis of the study is well motivated. The topic is timely. The manuscript is very well written. The results are robust and properly tested. However, I cannot recommend the study for publication because the conclusions of the study are not adequately supported by the experimental findings:

1. The findings are taken to support a neural hypothesis concerning the mechanism underlying schizophrenia concerning the strength of ascending and descending fibers. However, the findings are exclusively behavioral, no neural data is presented, and no neural manipulation is employed. Such measures would be required for drawing conclusion about neural connections.

2. It is unclear whether the best-fitting model is in any way different from (and superior to) any generic model that allows overweighing of evidence, since neither of the alternative models to which the best-fitting model was compared allows overweighing. If the answer is no, then a simpler model would likely suffice and the results and conclusions need to be reframed in terms of weighing of evidence rather than in terms of neural connections. The speculative link with neural connections can then be made in the introduction/discussion. Alternatively, if the answer is yes, a more extensive model comparison procedure is required to compare the best-fitting model to various other models that allow overweighing of evidence/priors. However, my impression is that the data set is not rich enough to support such an analysis.

3. The modeling results were obtained by excluding the trials in which sensory evidence was unambiguous. This is unsatisfactory because it is exactly in those trials where schizophrenia patients displayed behavior that is not explained by the best-fitting model (lower confidence than

the control group).

Minor comments:

4. Figure 1c: the line shades are hard to distinguish.
5. Figure 2a: explain why the spacing between the lines is not linear.
6. Figure 2c: descending loop arrow should be thicker like in D.
7. Page 9: the use of the word "inferior" is confusing because it implies poorer performance.
8. Page 11: It seems that the correlation with WM was tested across the whole sample (rather than only within groups) only for w_p and not for the other parameters. Is this true, and if it is, why?
9. Page 21: Explain why the reverberating terms appear twice each and not just once each.
10. Page 21-22: "Trials with non-informative sensory evidence (i.e., a log likelihood ratio=0) were excluded from the analysis to counterbalance the log prior ratio with the log likelihood ratio." I am not sure I understand why these needed to be counterbalanced.

Reviewer #3 (Remarks to the Author):

Jardri et al. "Sense what you expect or expect what you sense? Experimental evidence for circular inference in schizophrenia"

I very much enjoyed reading this excellent paper which is very well written and rests on a compelling modelling approach. I also liked the variation of the Fisher task you developed and think this has the potential for developing into a standard paradigms for probing probabilistic inference in schizophrenia. I have a few comments and suggestions which I hope you will find useful for a revision of your paper.

Major points

1. I will begin with one of my standard questions: Will the data and the code of the model be made freely available?
2. My main concerns and suggestion concerns the methodology of your group-level model

comparison. You compute a group-level BIC score, based on averaging the model's prediction across all subjects. This corresponds to a fixed effects approach which assumes that all subjects use the same model (and any variation originates from observation noise). This has two serious drawbacks: first, this approach is highly vulnerable to single outlier subjects. Second, the validity of the fixed effects is questionable for almost any cognitive paradigm; in your particular case, it seems particularly unlikely given large inter-individual variability in behaviour, both within the control group and in the patient group. I think it would be important to redo the model comparison and employ a random effects procedure, for example, as described by Stephan et al. (NeuroImage 2009, 46: 1004-1017). This approach treats the model as a random variable, allows one to quantify the degree of heterogeneity in a group, and is extremely robust against the influence of outliers on group results. Running this random effects BMS procedure is straightforward and does not mean much work: you can simply feed your matrix of BIC scores (subjects x models) into the `spm_bms` function from the open source code package SPM.

3. For those not intimately familiar with your previous work on circular inference, it would be helpful if you could add a slightly more detailed summary of the main ideas of this hypothesis. In particular, it would be helpful if you could explain how exactly an imbalance in excitation/inhibition would lead to reverberatory message passing. I say this because in other hierarchical Bayesian inference schemes top-down predictions exert an inhibitory effect on the next lower level (e.g., glutamatergic top-down predictions targeting GABAergic interneurons), effectively implementing a subtraction of the prediction from the sensory input which enters from the level below. Under this view, a disturbance in E/I balance would simply lead to an aberrant prediction error signal, but not necessarily in reverberation.

Minor points

4. Your introduction to schizophrenia (page 2) portrays the attempts of defining subgroups or subdimensions as if a mechanistically distinct and homogenous disease entity was broken up. I think this is potentially misleading: on the contrary, all evidence points to schizophrenia being a highly heterogeneous construct. I would recommend reformulating this introduction in order to avoid confusion in readers without a clinical background in psychiatry.

5. Page 9: you state that the BIC of the circular inference model was "largely inferior" - this formulation is misleading and will confuse readers. I think you need to point out here that in your use of the BIC, smaller values mean a better model and that the circular inference model was in fact superior to the simple Bayes model.

6. The implication of your modelling results that the controls displayed some degree of circular inference is extremely interesting. This may indicate that ongoing message passing is not necessarily pathological. See also the behaviour of predictive coding models in continuous time,

for example, Friston (Phil. Trans. R. Soc. B 2005, 360: 815-836) or the more recent formulation by Bogacz (2016, Journal of Mathematical Psychology, doi:10.1016/j.jmp.2015.11.003).

7. Several previous papers have discussed the possibility that a key pathophysiological feature in schizophrenia is that the weight or precision of sensory information is too high relative to the precision of prior beliefs (e.g., Adams et al. 2013, Corlett et al. 2010). This fits very nicely to your results from the parameter estimates of the circular inference model (page 10) and might be worth mentioning in the Discussion.

8. In Figures 2 and 3, could you please add colour bars so that one can directly relate the lines to different probabilities?

I hope you will find these thoughts and suggestions helpful and look forward to your response.

Klaas Enno Stephan

Reviewer #1.

C1. Thank you for sending me this paper for review. It presents a novel task, something of a mixture between random dots tasks and jumping to conclusions tasks, which examines how subjects integrate two very different types of information. The task, the analysis and the results are striking, very innovative and have the potential to have a strong impact on the field. The writing is clear and the paper a pleasure to read. I do have quite a few comments, and list them below. Although some of these read as very fundamental and hard comments, I would like to emphasize that, first, I think that all of these can be addressed with some additional analyses or extra reporting, and that even if some of them cannot, these are limitations that do not detract from the broader importance of reporting the task, and in some way just attest to the depth, strength and structure of the task, results and modelling. I should add that I will not agree re-review the paper unless the figures/legends/tables are included inline (including suppl. figures).

R1. We thank Reviewer 1 for these encouraging comments. Please note that we followed the submission process and recommendations to the authors regarding figures and supplementary material; however, we re-uploaded all of these figures to avoid the possibility of a corrupted file. We also directly provided a document with in-line figures in the Supplementary Material to the manuscript for readability.

C2. The prior manipulation is poorly explained - please provide more details, and include the exact instructions with translations as supplementary material. The use of a basket suggests that this is how many fish were caught from each of the two lakes, and it is not clear whether the single fish presented was then drawn from these baskets, or from the lakes directly. The methods imply that the baskets were used to indicate the fisher's preference, which might be more understandable.

R2. We thank Reviewer 1 for this comment. We clarified this point, notably that the fish caught by the fisher directly came from one of the 2 presented lakes and that the baskets were in fact used to indicate the fisher's preference. The participants were shown several example responses to ensure that they did not misinterpret the task (see also our reply to comment **C3**). This point was clarified in the Methods section.

As suggested, we also added the translated instructions used in the task to the Supplementary Material.

C3. To what extent could differences between the groups be due to differences in the understanding of the task instructions? What aspects of the data speak to this? For instance, if one were to choose subjects with comparable prior weight (which might be one, admittedly imperfect, proxy for the comprehension of the prior manipulation), would the remaining differences still hold?

R3. We thank Reviewer 1 for this comment. Please note that the participants were not selected for the current experiment based on comparable prior weights, and we would not have had enough participants to reach significance if we had restricted our analyses to participants with comparable prior weights. However, we ensured that all of the participants

understood the task through a training session that included 3 steps of incremental task difficulty: the prior independent of sensory evidence (step 1), sensory evidence independent of the prior (step 2), and a combination of the two (step 3) as it shows in the final “Fisher task”. Participants started the experiment only after this training was validated, and they had to repeat these preliminary steps when large errors were reported. The following points were clarified in the Methods section.

Finally, to address comment **C23** (reviewer 2), we discovered a subtle difference in the way that the two groups computed unambiguous trials (with infinite likelihood logits: all red fishes on one side and all black fishes on the other). Control (CTL) participants employed a “rule-based” strategy in those trials, choosing the extreme of the scale systematically (even when w_s was smaller than 1). In contrast, patients with schizophrenia (SCZ) used a probabilistic strategy that was represented by the *circular inference model*. We now indicate this difference in a sub-section of the **Supplementary Material**, titled “*The particular case of unambiguous trials*”, and we show the fit of the model in those trials in **Supplementary Figure 2**.

C4. Similarly, the scale on which the prior lives is poorly defined, and there may well be a strange function mapping the logits defined by the authors to the perceived logits by subjects. Could such a warp could possibly also provide some of the nonlinearity around zero?

R4. We now provide a better definition of the mapping between size and the logit model. Some of the deviations from linearity might be confounded with distortions in the perception of size or subjective interpretations. This possibility is now mentioned as a cautionary note in the manuscript (page 12, paragraph 3, line 1). Although we cannot completely rule out this alternative, we think that our interpretation of the **differences** between participants (as **differences** in confidence levels) is convincing. First, accounting for the results regarding size distortion would require extremely severe perceptual deficits than have not been reported in patients with SCZ or CTLs. Second, if participants applied an unknown non-linear mapping between the perceived and true logits, then we would still expect this mapping to be the same for the prior and likelihood information. This possibility is clearly not the case (e.g., α_p and α_s are completely uncorrelated).

C5. Patient description is too superficial. The average dose of medication is on the high side with 20mg of olanzapine, suggesting that this is quite a severely ill sample. Information about patient characteristics needs to be in the main text. Suppl. Information about patients needs to contain medication (not just olanzapine equivalents), medication history as far as known, and a characterisation of severity in terms of illness course: years of illness, occupational status and comorbidities, number of hospitalisations etc.

R5. We followed the reviewer’s recommendation and provided more information regarding the tested samples in Table 1, which now includes illness duration (mean duration in years+/- standard deviation [SD]), number of hospitalizations (mean+/-SD), and the number/ratio of patients unemployed or disabled (n and %). Furthermore, we moved **Table 1** to the main text as suggested.

C6. The notion of 'reverberating loop' is strange, as they're not really loops, but two separate factors, one for the prior and one for the evidence. It would be helpful to clarify this in terms of the language and interpretation. This might be a misunderstanding that could be helped by providing the derivations for the reverberating model.

R6. We thank Reviewer 1 for indicating this potential source of confusion. In fact, we have **separate** parameters: α_p for "descending loops" and α_s for "ascending loops". α_s and α_p are uncorrelated and are related to different clinical dimensions in SCZ, suggesting that they are largely independent processes.

However, the prior and likelihood information **strongly interact** in the *circular inference models* because of their appearance in both the bottom-up factor $F(\dots, w_s)$ and the top-down factor $F(\dots, w_p)$. This result contrasts with the two independent prior and likelihood factors in the weighted Bayesian model. This interaction reflects the fact that reverberation inevitably causes both types of information to be corrupted by the other.

We clarified this point by providing more details regarding how we derived the components of the model in the Methods section. We now also indicate the interaction between the prior and likelihood information present in the data in the Results section (bottom of page 9, top of page 10).

C7. Discussion: "in SCZ, tiny changes in the likelihood logit around 0 could cause extremely large changes in the chance logit" This is cryptic for: when the fish proportions differed only slightly, patients nevertheless showed strong differences, I assume? Given that they did not attend much to the prior, it seems that in this difficult problems, something particular does in fact happen. Is there any corroborating evidence about what subjects did e.g. from debriefing? Could this be not an effect of likelihood, but an effect of "difficulty"?

R7. We supplemented this "cryptic" description with a more concrete description of the effects following your suggestion. Please note that these large slopes around likelihood logit 0 occur even among patients who accounted for the prior, as confirmed by an absence of a significant correlation between α_s and w_p or α_p . Moreover, although the reaction times (RTs) of patients with SCZ were slower, we did not find a correlation between RT (which might reflect task difficulty) and any of the model parameters, including α_s .

C8. It is a shame that the reaction times have not been examined. Would the authors not have expected them to correlate with the amount of 'reverberation', may be $\alpha_p + \alpha_s$?

R8. Our version of the model does not include time. In a recurrent network, longer processing time might be associated with more reverberations. However, we found no significant correlations between RT and the model parameters. The only significant relationship we found was that briefer RTs were associated with less ambiguous likelihood (see **Figure 1b**; the statistics are reported in the main text). This finding was expected because visually discriminating between slightly different fish proportions is more difficult than visually discriminating between very different ones. To observe any relationship between circular inference and RT, we would most likely need an experiment in which sensory processing difficulties and RTs were carefully controlled.

C9. Discussion: "The most salient result of this study is the large over-counting of sensory evidence in schizophrenia" I don't understand this conclusion – the group effect on the prior is equally large and more significant.

R9. You are right. We now mention that two salient results are present: one, the over-counting of sensory evidence relative to positive symptoms; and two, the under-weighting of the prior linked to working memory (WM) deficits.

C10. In the results on correlation with PANSS, there are a number of statements 'correlated with x, but not with y'. These should be supported by direct test using interactions.

R10. We thank Reviewer 1 for this comment. We now report significant interactions to support our initial claims (a statistical test regarding the difference rather than 2 separate tests; see also our reply to comment **C18**).

C11. Minor point. The distinction between prior and evidence is somewhat fuzzy. Could one equally well turn around the two, call 'evidence' the size of the basket, and prior the proportions in the lake. What would the consequences be for the model / interpretation of the results?

R11. The presence of stronger circular inference loops in patients with SCZ does not depend on such distinctions or the widely different amounts of loops for the two types of information. However, our interpretation (in terms of ascending and descending loops) depends on it.

We interpreted the basket information as a prior because (i) it preceded the lakes and was removed during the response stage in contrast with fish colour information; (ii) the basket sizes were not directly informative about the fish colour; and (iii) participants were specifically instructed to interpret basket size as a preference of the fisher for one of the lakes. Beyond the terminology (which can always be debated), what appears important for our interpretation is whether these two types of information primarily recruited the feed-forward processing of sensory information or a top-down prediction based on prior knowledge. Therefore, we interpreted the memorized basket information as primarily "top down" and the processing of the fish proportion/colours as primarily "bottom up".

C12. Minor point. I don't understand the F tests: the values look like they were performed as fixed effects, but the methods state they were mixed effects. Could you please clarify how the degrees of freedom arise?

R12. In fact, we used a *general linear mixed model* to model the parametric effects of sensory evidence, prior congruency and their interaction as fixed effects with repeated measures across participants, and participants were treated as a random factor. The full analyses also modelled the fixed effects of the group without repeated measures across participants. We modelled the variance-covariance matrix of the fixed effects with a simple variance component (i.e., diagonal structure) to estimate the contribution of the random effect to the variance of the fixed effects, without additional assumptions regarding the contribution to the covariation between the fixed effects. We rephrased the Methods section to clarify this point.

The denominator degrees of freedom were computed using the Welch–Satterthwaite equation. This equation approximates the effective degrees of freedom of a linear combination of independent sample variance. For clarity, we report the denominator degrees of freedom rounded to the nearest integer; however, we emphasize that they are actually non-integers because these statistics do not have exact F-distributions.

C13. Minor point. p6: I don't understand how the lines in the figures, which seem to show rather striking differences in the effects of congruency between the groups is compatible with "on the remaining trials, prior congruency modulated the effect of sensory evidence similarly in the two groups (three-way interaction: $p=0.80$ ", particularly in the view of the significance of visually far smaller effects, e.g. the impact of prior congruency per se in the SZ group.

R13. We acknowledge that the description of the interactions was not clear enough. We rephrased most of these descriptions in the new version of the manuscript. First, we describe the two-way interaction between *sensory evidence x prior congruency* in the CTLs by providing breakdown analyses across trial types to show how the prior influenced the confidence rate with regard to each type of sensory evidence. Next, we describe the significant three-way interaction among *group x sensory evidence x prior congruency* using similar breakdown analyses to show how the differential influence of the prior varied between the two groups across trial types.

C14. Minor point. The model assumes that the information extracted from both visual stimuli is the same as that extracted from one. Can this assumption be substantiated?

R14. The task was only built to collect one response for the 2 lakes presented simultaneously. Thus, a symmetrical proportion of fishes appeared in the 2 lakes to simplify the task complexity.

C15. Minor point. Please provide a derivation of the recursive model - it is not obvious to me exactly how this is arrived at.

R15. Our equation sought to capture the general effects of circular inference in a network (i.e., the over-counting of the same information, the corruption of sensory evidence by the prior and vice versa) using the simplest possible model and a minimal number of free parameters. This equation is not a direct mathematical derivation of the output of a recurrent network with a circular inference; such a model has no analytical expression and requires simulations. Furthermore, it would contain far too many parameters/degrees of freedom (i.e., numbers of iterations, numbers of layers, synchronous versus asynchronous updates of the beliefs, and so on). To clarify this point, we now provide a more detailed justification for the different model components in the Methods section.

C16. Minor points. Please superimpose the model fits on the data in figure 1b,c. This would clearly show how good a model this is, and that it can capture ever strikingly complex and counterintuitive aspects of the data.

R16. We updated **Figure 1** accordingly. Now, we used a colour-coded scheme similar to what was proposed in Figure 3 to ensure readability (see also comment **C24**). Model predictions in unambiguous trials are not presented in Figure 1 for the reasons presented in **R3** and discussed in the **Supplementary Material**.

C17. Minor point. Parameter estimates, p10: "In the CTLs, the mean sensory weight () was close to 1 (Fig. 5a), but the prior weight () was less than 1." -> this is not compatible with the figure, which shows 0.6 and 0.5, respectively!

R17. We have changed the text accordingly:

*"The mean sensory weight (w_s) and prior weight (w_p) of the CTLs were 0.64 and 0.56, respectively. This result is not surprising given the qualitative nature of the basket size information (see **Methods**). The smaller sensory weights might reflect uncertainty about the exact fish proportion in the two lakes."*

C18. Minor point. "However, WM did not differentially affect the other parameters values within the SCZ and CTL groups ($p=0.24$)." why is there only one p value for all other parameters for both groups? How exactly was this tested?

R18. We apologize for this omission. In accordance with the recommendation made in **C10**, we now report a significant interaction to support our initial claim.

C19. Minor point. Please label the axes in 5b,c, and 6.

R19. **Figures 5 and 6** have been updated accordingly.

C20. Minor point. Language: 'fishes distribution' -> 'fish distribution' or 'distribution of fishes'/'severity of ascending loops' -> disorders have a severity, but not loops. maybe strength?

R20. We made these necessary changes and used a professional English editing service before resubmitting our manuscript.

Reviewer #2.

C21. The findings are taken to support a neural hypothesis concerning the mechanism underlying schizophrenia concerning the strength of ascending and descending fibers. However, the findings are exclusively behavioral, no neural data is presented, and no neural manipulation is employed. Such measures would be required for drawing conclusion about neural connections.

R21. Our *circular inference model* was derived from a neural hypothesis; thus, we cannot completely remove the focus on the excitatory/inhibitory (E/I) balance. However, to

emphasize the fact that our results do not provide any direct neurophysiological evidence, we rearranged the summary such that the E/I balance appears as a possible interpretation of the results and does not represent the main conclusion of the study.

C22. It is unclear whether the best-fitting model is in any way different from (and superior to) any generic model that allows overweighing of evidence, since neither of the alternative models to which the best-fitting model was compared allows overweighing. If the answer is no, then a simpler model would likely suffice and the results and conclusions need to be reframed in terms of weighing of evidence rather than in terms of neural connections. The speculative link with neural connections can then be made in the introduction/discussion. Alternatively, if the answer is yes, a more extensive model comparison procedure is required to compare the best-fitting model to various other models that allow overweighing of evidence/priors. However, my impression is that the data set is not rich enough to support such an analysis.

R22. We apologize for the lengthy answer, but it is important for us to distinguish our approach from other “generic” models of weighted-cue combination. The short answer is that we used binary choices not continuous variables, such as position or direction of motion. Thus, a weighted-cue combination model is not appropriate and would provide an extremely poor fit to the data (i.e., a “straw man”). We provide more details below.

The generic model enables an overweighing of sensory evidence or prior estimates (including free energy models and predictive coding models) are applicable to continuous variables. These models are “exact” only when the variables are distributed normally. Its direct application to binary variables, such as a choice between the right and left lakes, would provide absurd predictions. For example, this would lead to a confidence estimate of “2” when both the prior and likelihood unambiguously favour the right lake (i.e., their estimate is 1).

In our case, the closest equivalent to the continuous case scenario is the *weighted Bayes model* (i.e., a Bayesian model in which the assumed precision of the sensory and prior information is allowed to vary). This information has been clarified in the Results section. We also added a paragraph to the Discussion section to explain these differences with weighted-cue combination approaches.

If the reviewer was referring to a weighted sum of logits, then this is a special case of the *circular inference model* in which the sensory evidence and prior are completely trusted (i.e., $w_p=w_s=1$). In that case, however, the weights do not represent an assumed reliability; rather, they correspond to amount of loops α_s and α_p (plus 1).

C23. The modeling results were obtained by excluding the trials in which sensory evidence was unambiguous. This is unsatisfactory because it is exactly in those trials where schizophrenia patients displayed behavior that is not explained by the best-fitting model (lower confidence than the control group).

R23. We thank the reviewer for pointing out this problem. We should consider how the model performs with regard to those trials in light of results shown on **Figure 1**.

We included “unambiguous” trials as an experimental control to ensure that the participants used the entirety of the scale. If all of the fishes were red in one of the lakes but black in the other, then it would be impossible for a black fish to originate from the former lake (note that likelihood logits are infinite in those cases). In addition, the participants were instructed to click on the extremity of the scale in those trials (see the translated instructions in Supplementary Material). We expected that all participants would use a purely rule-based strategy for those unambiguous trials (see also our reply to comment **C3**, Reviewer 1).

The above scenario was clearly the case for CTLs. These participants systematically clicked on the extremity of the scale during unambiguous trials, regardless of how they weighed the sensory evidence and during prior ambiguous trials (e.g., their best fitting w_s and w_p). If we apply the model to the unambiguous trials not used to fit the model, it systematically underestimates their confidence level (see **Figure 1**).

Interestingly, this event did not occur to the same extent among patients with SCZ. The model is able to predict their relatively low confidence level and the fact that they continue to be influenced by the prior (as long as $w_s < 1$ and $w_s > 0$).

To indicate this between-group difference, we now show the model predictions for unambiguous trials in **Figure 1b** and **Supplementary Figure 2**, and comment on it in a subsection of the *Supplementary Material*.

To completely address your concern regarding the robustness of our results, we also fit the model to **all** of the trials, bounding the otherwise infinite likelihood logits between -3 and 3. As expected, the model provides a poor fit to unambiguous trials among CTLs, continuing to underestimate their confidence level (albeit to a lesser extent). As a compromise, it overestimates the confidence level for ambiguous trials. Nevertheless, none of our main statistical results changed: The *circular inference model* continues to outperform the weighted and *simple Bayes model* in terms of Bayesian information criterion (BIC) scores; furthermore, α_s is significantly larger in patients with SCZ, α_p is correlated with negative symptoms, α_s and $\alpha_s + \alpha_p$ are correlated with dissociative symptoms. The parameters fit on the whole dataset are highly correlated with the parameters fit on only the ambiguous trials.

C24. Minor point. Figure 1c: the line shades are hard to distinguish. Figure 2a: explain why the spacing between the lines is not linear. Figure 2c: descending loop arrow should be thicker like in D.

R24. **Figure 1c** has been updated based on comment **C16**, and we changed the greyscale to a colour image to improve readability. Regarding **Figure 2**, colours were used to correspond to probabilities (0.1, 0.2, and so on) while the scale is in logits. This change explains why the spaces between the curves are non-linear. Colour bars were added to the figures for clarification. Furthermore, the size of arrows in the graph models directly represents the weights w_p and w_s . In the examples displayed in Figures **2c** and **2d**, $w_s > w_p$ (0.9 and 0.75, respectively), and feed-forward messages are thicker than feedback messages, independent of α_s or α_p (which results in loops). To avoid confusion, we also clarified the loops as curved arrows to differentiate them from the initial weights.

C25. Minor point. Page 9: the use of the word "inferior" is confusing because it implies poorer performance.

R25. We thank Reviewer 2 for this comment and agree that this formulation might have been misleading (see also our reply to comment **C33**). The text has been modified as follows:

"Smaller BIC values denote a better fit. Over all the participants, the BIC of the circular inference model (1,220) was largely inferior to that of the simple Bayes model (8,464) and the weighted Bayes model (5,293), confirming the better fit of the circular inference model over the two other models."

C26 Minor point. Page 11: It seems that the correlation with WM was tested across the whole sample (rather than only within groups) only for w_p and not for the other parameters. Is this true, and if it is, why?

R26. We thank Reviewer 2 for this comment. Of course, we tested for the various possible associations between the model parameters and WM performance, but only the WM/ w_p correlation was significant. According to Neuenhuis et al., 2011 and following the recommendations of Reviewer 1 (see our reply to comment **C10**), we now report the significant interactions to support our initial claims via a statistical test concerning the difference, rather than separate correlation tests.

C27 Minor point. Page 21: Explain why the reverberating terms appear twice each and not just once each.

R27. In a hierarchical inference system, top-down and bottom-up information should be treated independently. However, any reverberation up and down the hierarchy would destroy this independence, causing both the prior information (i.e., the top-down factor) and sensory evidence (i.e., the bottom-up factor) to be equally corrupted by over-counted likelihoods and priors. Thus, the reverberating terms appear twice, once in the top-down factor and once in the bottom-up factor. We clarified this point by providing an additional explanation for the model (see also our reply to comment **C6**, Reviewer 1).

C28. Minor point. Page 21-22: "Trials with non-informative sensory evidence (i.e., a log likelihood ratio=0) were excluded from the analysis to counterbalance the log prior ratio with the log likelihood ratio." I am not sure I understand why these needed to be counterbalanced.

R28. We apologize for this mistake in reporting the procedure. Trials with non-informative sensory evidence were not excluded from the analyses.

Reviewer #3.

C29. I very much enjoyed reading this excellent paper which is very well written and rests on a compelling modelling approach. I also liked the variation of the Fisher task you developed and think this has the potential for developing into a standard paradigms for probing probabilistic inference in schizophrenia. I have a few comments and suggestions which I hope you will find useful for a revision of your paper. I will begin with one of my standard questions: Will the data and the code of the model be made freely available?

R29. We thank Reviewer 3 for this supportive comment. We will make the data available upon request to the corresponding author and directly provide the Matlab scripts for the model in the Supplementary Material. These points have been specified in both the manuscript checklist and the main text.

C30. My main concerns and suggestion concerns the methodology of your group-level model comparison. You compute a group-level BIC score, based on averaging the model's prediction across all subjects. This corresponds to a fixed effects approach which assumes that all subjects use the same model (and any variation originates from observation noise). This has two serious drawbacks: first, this approach is highly vulnerable to single outlier subjects. Second, the validity of the fixed effects is questionable for almost any cognitive paradigm; in your particular case, it seems particularly unlikely given large inter-individual variability in behaviour, both within the control group and in the patient group. I think it would be important to redo the model comparison and employ a random effects procedure, for example, as described by Stephan et al. (NeuroImage 2009, 46: 1004-1017). This approach treats the model as a random variable, allows one to quantify the degree of heterogeneity in a group, and is extremely robust against the influence of outliers on group results. Running this random effects BMS procedure is straightforward and does not mean much work: you can simply feed your matrix of BIC scores (subjects x models) into the `spm_bms` function from the open source code package SPM.

R30. We thank Reviewer 3 for this excellent suggestion. We applied the `spm_bms` function to the individual BIC scores in the three models and now report the results together with the BIC scores. The conclusion stays the same: The *circular inference model* shows a strong dominance.

Please note that for the reported BIC, we accounted for the fact that different participants have different fit parameters, e.g., the number of free parameter is 4 x the number participants for the *circular inference model*.

C31. For those not intimately familiar with your previous work on circular inference, it would be helpful if you could add a slightly more detailed summary of the main ideas of this hypothesis. In particular, it would be helpful if you could explain how exactly an imbalance in excitation/inhibition would lead to reverberatory message passing. I say this because in other hierarchical Bayesian inference schemes top-down predictions exert an inhibitory effect on the next lower level (e.g., glutamatergic top-down predictions targeting GABAergic interneurons), effectively implementing a subtraction of the prediction from the sensory input which enters from the level below. Under this view, a disturbance in E/I balance would simply lead to an aberrant prediction error signal, but not necessarily in reverberation.

R31. We thank Reviewer 3 for this important comment. We addressed this point by clarifying the mechanisms that lead to reverberation due to impaired inhibition and the difference with predictive coding in the introduction (page 2, paragraph 2). Notably, we based this argumentation on two recent reviews/opinion papers (Denève & Jardri, *Curr Opin Behav Sci* 2016 and Jardri et al., *Schizophr Bull*, in press), in which we provide a much more detailed comparison using predictive coding.

C32. Minor point. Your introduction to schizophrenia (page 2) portrays the attempts of defining subgroups or subdimensions as if a mechanistically distinct and homogenous disease entity was broken up. I think this is potentially misleading: on the contrary, all evidence points to schizophrenia being a highly heterogeneous construct. I would recommend reformulating this introduction in order to avoid confusion in readers without a clinical background in psychiatry.

R32. We thank Reviewer 3 for this comment and altered the text in the first paragraph to follow this suggestion, insisting on heterogeneity and citing the equifinality model for SCZ.

C33. Minor point. Page 9: you state that the BIC of the circular inference model was "largely inferior" - this formulation is misleading and will confuse readers. I think you need to point out here that in your use of the BIC, smaller values mean a better model and that the circular inference model was in fact superior to the simple Bayes model.

R33. We thank Reviewer 3 for this comment. We agree that this formulation might be misleading and modified it accordingly (see also our reply to comment **C25**).

C34. Minor point. The implication of your modelling results that the controls displayed some degree of circular inference is extremely interesting. This may indicate that ongoing message passing is not necessarily pathological. See also the behaviour of predictive coding models in continuous time, for example, Friston (Phil. Trans. R. Soc. B 2005, 360: 815-836) or the more recent formulation by Bogacz (2016, Journal of Mathematical Psychology, doi:10.1016/j.jmp.2015.11.003).

R34. We thank Reviewer 3 for this comment and have added these references in support of the discussion regarding minimal circular inference in CTLs.

C35. Minor point. Several previous papers have discussed the possibility that a key pathophysiological feature in schizophrenia is that the weight or precision of sensory information is too high relative to the precision of prior beliefs (e.g., Adams et al. 2013, Corlett et al. 2010). This fits very nicely to your results from the parameter estimates of the circular inference model (page 10) and might be worth mentioning in the Discussion.

R35. We thank Reviewer 3 for this comment and have now included these references.

C36. Minor point. In Figures 2 and 3, could you please add colour bars so that one can directly relate the lines to different probabilities?

R36. We updated **Figures 2** and **3** accordingly.

Reviewers' Comments:

Reviewer #1 (Remarks to the Author):

All my comments have been satisfactorily addressed. I would like to congratulate the authors on an excellent and important study.

Reviewer #2 (Remarks to the Author):

The authors now clarify in the discussion that the main result is indicative of over-counting of sensory evidence in Schizophrenia, and the 'ascending/descending loops' perspective is only one possible interpretation. This result is well supported by the data and I find it to be of interest to Schizophrenia research. However, most of the text and especially the abstract and the title still present the 'ascending/descending loops' interpretation as the only natural interpretation of the results. Personally, I am not even convinced that this is the most likely interpretation. The problem is two fold:

1. Is it necessary to invoke corruption of sensory evidence by prior and vice versa to explain subjects' behavior? Additional analyses could clarify this. Specifically, it is possible to test the data against two additional models that allow over-counting but do not involve any reverberating mechanism: one would simply be a simplification of the circular inference model that allows over-counting but not 'reverberation' (the equation would become: $L_c = F(L_s + F(A_s L_s, W_s)) + F(L_p + F(A_p L_p, W_p))$). The other model would involve augmenting the weighted Bayes model with a non-linear transformation of the evidence and prior (i.e., a single-parameter sigmoid for S_r and another such function for P_r that allow these quantities to be amplified or weakened before they enter into the prior and likelihood computations). To argue that sensory evidence and prior information corrupt one another, it seems necessary that the circular inference model would explain the data better than these two alternative models. Otherwise, the results suggest a completely different neural model.

2. If the answer to (1) is positive, could such interaction between prior and evidence be explained in other ways that do not involve ascending/descending loops? For instance, jumping to conclusions is often explained by attractor dynamics in a network in which decision units compete through mutual inhibition. Perhaps the experimental results could be explained as well by assuming that in schizophrenia patients such attractor dynamics are particularly strong in circuits that relay sensory evidence.

Reviewer #3 (Remarks to the Author):

Thank you very much for your response. Your revisions are very helpful, and I was particularly reassured by the random effects BMS demonstrating that your results are not due to individual outliers.

There are only two minor issues left that do not require any further review and whose treatment I leave to your discretion.

First, concerning my previous point C31, the physiological motivation of your model would benefit from a more explicit explanation of the physiological evidence that a sensory input can indeed "reverberate" in cortical circuits.

Second, concerning my previous point C33, I think there is still a language issue: "inferior BIC" would not be understood by most readers as you expect, i.e. as a numerically lower value, but as a "worse model"; furthermore, "better fit" is an unfortunate term since BIC concerns the balance between fit and accuracy of a model. I would recommend reformulating the respective sentence as:

"Over all the participants, the BIC of the circular inference model (1,220) was considerably smaller than that of the simple Bayes model (8,464) and the weighted Bayes model (5,293), confirming that the circular inference model provided a better explanation of the data than the two other models."

I hope these additional (and minor) observations are helpful and would like to congratulate you on a very interesting and stimulating paper.

Klaas Enno Stephan

Reviewer #1 (Remarks to the Author)

Comment C#1. All my comments have been satisfactorily addressed. I would like to congratulate the authors on an excellent and important study.

R1. We would like to thank reviewer 1 for her/his encouragements and supports.

Reviewer #2 (Remarks to the Author)

Comment C#2. The authors now clarify in the discussion that the main result is indicative of over-counting of sensory evidence in Schizophrenia, and the 'ascending/descending loops' perspective is only one possible interpretation. This result is well supported by the data and I find it to be of interest to Schizophrenia research. However, most of the text and especially the abstract and the title still present the 'ascending/descending loops' interpretation as the only natural interpretation of the results. Personally, I am not even convinced that this is the most likely interpretation. The problem is two fold:

1. Is it necessary to invoke corruption of sensory evidence by prior and vice versa to explain subjects' behavior? Additional analyses could clarify this. Specifically, it is possible to test the data against two additional models that allow over-counting but do not involve any reverberating mechanism: one would simply be a simplification of the circular inference model that allows over-counting but not 'reverberation' (the equation would become: $L_c = F(L_s + F(A_s L_s, W_s)) + F(L_p + F(A_p L_p, W_p))$). The other model would involve augmenting the weighted Bayes model with a non-linear transformation of the evidence and prior (i.e., a single-parameter sigmoid for S_r and another such function for P_r that allow these quantities to be amplified or weakened before they enter into the prior and likelihood computations). To argue that sensory evidence and prior information corrupt one another, it seems necessary that the circular inference model would explain the data better than these two alternative models. Otherwise, the results suggest a completely different neural model.

2. If the answer to (1) is positive, could such interaction between prior and evidence be explained in other ways that do not involve ascending/descending loops? For instance, jumping to conclusions is often explained by attractor dynamics in a network in which decision units compete through mutual inhibition. Perhaps the experimental results could be explained as well by assuming that in schizophrenia patients such attractor dynamics are particularly strong in circuits that relay sensory evidence.

R2. We thank reviewer 2 for these comments.

Regarding the first point and as suggested, we tested the "no-reverberation" model, $L_c = F(L_s + F(A_s L_s, W_s)) + F(L_p + F(A_p L_p, W_p))$. Despite having the same number of free parameters, its performance in terms of BIC score is markedly weaker than the model with reverberation, i.e., "circular inference" ($BIC_{\text{reverberation}} - BIC_{\text{no reverberation}} = -95$).

This "no-reverberation" model was in fact the first model we thought of applying to the data (because of its simplicity). However, we rejected it early on since it predicts that the subject's confidence is the sum of two independent functions, one depending purely on the likelihood, the other depending only on the prior. In particular, this model predicts that the slopes of all the sigmoids in **Figure 3** and all the sigmoids in **supplementary Figures 3 and 4** would be identical. This is clearly not the case, i.e., the confidence is influenced more strongly by the likelihood when the prior is non-informative ($L_p = 0$) than when the prior is strongly positive or negative. Vice-versa, the prior influences more the confidence when L_s is small. Note that while the effect appears moderate in **Figure 3a,b** as a result of averaging heterogeneous subject data, it is much more evident in individual subjects (see **Figure 3c,d,e,f**, and **supplementary Figures**).

As regard to your second suggestion (augmenting the weighted Bayes model with a non-linear transformation of the evidence and prior), as far as we understand you mean something like " $F(\text{sigmoid}(L_s), w_s) + F(\text{sigmoid}(L_p), w_p)$ ". This alternative is also a sum of separate functions of

likelihood and prior, and suffers from the same shortcomings. Moreover, mathematically speaking, it is quasi-equivalent to the "no-reverberation" model. $F(\alpha L, w)$ is indeed a sigmoid with slope determined by α , and saturation level determined by w . Both are required to provide a good fit to the data.

Based on these considerations, we suggest to keep the prediction and result sections as simple as possible, and not confuse the reader with yet another model. However, we can provide the results of the model without reverberation in supplementary information if it is deemed necessary.

Regarding the second point on "mutual inhibition", this type of models accounts for mechanisms of perception and action selection in discrimination tasks that involve selecting a correct response and inhibiting an incorrect response. This predicts binary choices, not graded responses about confidence as required in this experiment. Nonetheless, we acknowledge that this could account for the response of the patients with the most extreme over-counting. In contrast, many patients, and all controls have more moderate effects and provide graded confidence levels on a trial by trial basis. Maybe we could argue that the attractor model could be modified to provide graded responses, but what would be the point? The Circular Inference model as implemented and tested in this paper is a normative account, predicting behavior accurately and in a parsimoniously fashion, while being generalizable to other inference tasks. We do not think that a mathematically equivalent mechanistic model, such as an attractor, would bring additional insights.

Reviewer #3 (Remarks to the Author)

Comment C3. Thank you very much for your response. Your revisions are very helpful, and I was particularly reassured by the random effects BMS demonstrating that your results are not due to individual outliers. There are only two minor issues left that do not require any further review and whose treatment I leave to your discretion. First, concerning my previous point C31, the physiological motivation of your model would benefit from a more explicit explanation of the physiological evidence that a sensory input can indeed "reverberate" in cortical circuits.

R3. We would like to thank reviewer 3 for this suggestion, and have added the following sentence in support for a possible reverberation of sensory inputs in cortical circuits:

"This finding appears compatible with recent physiological recordings coupled with optogenetic inhibition which revealed that reverberation, with notably recurrent inputs to sensory areas, is essential for accurate perception (Manita et al., Neuron 2015)."

Comment C4. Second, concerning my previous point C33, I think there is still a language issue: "inferior BIC" would not be understood by most readers as you expect, i.e. as a numerically lower value, but as a "worse model"; furthermore, "better fit" is an unfortunate term since BIC concerns the balance between fit and accuracy of a model. I would recommend reformulating the respective sentence as:

"Over all the participants, the BIC of the circular inference model (1,220) was considerably smaller than that of the simple Bayes model (8,464) and the weighted Bayes model (5,293), confirming that the circular inference model provided a better explanation of the data than the two other models."

I hope these additional (and minor) observations are helpful and would like to congratulate you on a very interesting and stimulating paper.

R4. We thank reviewer 3 and have altered the text as suggested in the revised version of the paper.

Reviewers' Comments:

Reviewer #2 (Remarks to the Author):

I thank the authors for their response. I find their answers satisfactory, and I believe that the readers would benefit from having the same information as well. In particular, it is important to mention in the discussion that it may be possible to formulate other mechanistic accounts of the results that do not involve ascending/descending loops (e.g., via attractor dynamics). Second, it makes sense to inform the readers that the no-reverberation model was tested as well, at least by mentioning it in the methods section.